# ON TRAINING MIXTURE-OF-EXPERTS: A SOCIAL CHOICE PERSPECTIVE

## ABSTRACT

Mixture-of-Experts (MoE) training faces a dilemma between expert specialization and balanced computation. We recast this problem through the lens of social choice theory, attributing training difficulties to Arrow's Impossibility Theorem. Inspired by this, we propose Regulated Mixture-of-Experts (RMoE), comprising a phased curriculum for load-balancing and stateful fusion for expert weighting. Experiments on GLUE and DomainBed show RMoE significantly outperforms standard MoE and dynamic routing baselines. Furthermore, RMoE demonstrates strong scalability on large-scale reasoning tasks with Qwen3 and Mixtral architectures. Our code is available at https://anonymous.4open.science/r/R-MoE-E3DC.

## 1 INTRODUCTION

The Mixture-of-Experts (MoE) architecture has emerged as a dominant paradigm for efficiently scaling the capacity of large models (Shazeer et al., 2017; Lepikhin et al., 2020). By replacing dense feed-forward network (FFN) layers with a set of smaller "expert" sub-networks, MoE models can possess trillions of parameters while keeping the computational cost per input constant. Despite its appealing, training MoE often leads to "routing collapse," where the router disproportionately sends most tokens to a few "winning" experts, leaving others under-trained and wasting model capacity (Shazeer et al., 2017; Lepikhin et al., 2020). The standard solution is to add an auxiliary load-balancing loss to the primary task objective, encouraging a more uniform distribution of tokens across experts (Fedus et al., 2022). This, however, leads to the difficult trade-off between the primary task and loading balancing objectives (Wang et al., 2024).

In this paper, we view MoE routing from a novel perspective through the lens of social choice theory (Sen, 1977; 1986). Social choice theory, a foundational framework in economics and political science (Little, 1952; De Condorcet et al., 2014), addresses dilemmas in group decision-making where agents must select from alternatives while balancing efficiency (maximizing overall utility) and fairness (ensuring equitable representation) (Rawls, 2017; Sen, 2017; Moulin, 1991; Gibbard, 1973; Satterthwaite, 1975). We reframe MoE routing as a problem of achieving agreement in a multi-agent system where each token (i.e., agent) needs to select a "committee" (i.e., expert) ( §2). . Under this perspective, the primary task loss and auxiliary load balancing loss are considered as efficiency and fairness, respectively, and we argue that the difficult trade-off in training MoE between efficiency and fairness is attributed to the Arrow's impossibility theorem (Black, 1969; Kelly, 2014). Taking inspirations from prior studies in social choice theory (Dreze, 1985; Pitis & Zhang, 2020; Caragiannis et al., 2023), we then propose a novel framework, RMoE, to alleviate the difficult training in MoE (§3). Specifically, RMoE consists of two strategies, which are called Phased Curriculum and Stateful Fusion. In phased curriculum strategy, inspired by the Second-Best Theory (Dreze, 1985; Ben-Yashar & Milchtaich, 2007), we optimize the efficiency at the early of training and then gradually bias to the fairness through a time-varying dynamic interpolation coefficient. In Stateful Fusion strategy, we incorporate the prior knowledge into the decision function such that two adjacent tokens in a training instance may have dependent expert distributions because of their dependency in semantics.

We demonstrate the effectiveness of our unified framework through extensive experiments (§4). Our model achieves state-of-the-art results on several GLUE benchmark (Wang et al., 2018) tasks and significantly improves the performance of Vision Transformer (ViT (Dosovitskiy et al., 2020)) MoE models on challenging domain generalization benchmarks like PACS (Li et al., 2017), VLCS

(Albuquerque et al., 2019), OfficeHome (Venkateswara et al., 2017), and DomainNet (Peng et al., 2019). These results confirm that regulating the training process provides a more effective path to training MoE models.

Our contributions are:

1. **A Novel Perspective on MoE Training:** We introduce a connection between MoE routing and committee selection from social choice theory. This provides a new perspective to understand the challenges in MoE training.

2. **A New Approach on MoE Training:** Under the social choice perspective, we propose a new approach RMoE to training MoE, consisting of Phased Curriculum and Stateful Fusion, which leads to better performance than strong baselines.

## 2 RELATED WORK

**Load Balancing in MoE.** MoE training focuses on balancing expert utilization to prevent routing collapse. The standard approach, as in Switch Transformer (Fedus et al., 2022), uses a static auxiliary loss penalizing imbalance, but its fixed weight causes gradient interference: $\nabla_{\theta_g} L = \nabla_{\theta_g} L_{task} + \alpha \nabla_{\theta_g} L_{aux}$, where components often conflict. Recent methods like DynMoE (Guo et al., 2025b) enable adaptive gating for dynamic expert activation, altering the computational graph. Loss-Free Balancing (Wang et al., 2024) eliminates the auxiliary loss by adding dynamic biases to router logits for reactive equilibrium without interference. Early efforts relied on static penalties (Jacobs et al., 1991; Jordan & Jacobs, 1994; Auda & Kamel, 1997; Eigen et al., 2013), evolving to adaptive mechanisms paralleling social reforms for fairness. Our curriculum schedules balancing intensity, prioritizing early differentiation and later stability under the hypothesis that training stages need varying regulation.

**Expert Fusion and Routing Mechanisms.** Standard MoE couples expert selection and weighting from instantaneous router scores. Recent works decouple them (Jiang et al., 2024). Soft MoE (Puigcerver et al., 2023) uses differentiable soft assignments, with experts processing weighted averages of tokens, avoiding sparse issues but shifting to dense mixing. MomentumSMoE (Teo & Nguyen, 2024) applies momentum to stabilize token representations across layers, smoothing update trajectories. Our stateful fusion is orthogonal: it applies momentum to fusion weights within a layer for stable expert outputs, motivated by our stability framework, unlike stabilizing inter-layer information flow.

**Social Choice Applications in AI.** Computational social choice combines theory with algorithms for aggregating preferences in multi-agent systems like voting and allocation (Brandt et al., 2016; Endriss, 2011). It addresses hardness with approximations and aligns AI with human values, viewing MoE routing as aggregating token preferences where imbalance mimics manipulation (Chevaleyre et al., 2007). Arrow's theorem extensions highlight limits, inspiring randomized or restricted solutions (Kelly, 2014). This frames MoE as sequential dilemmas with path dependencies; our RMoE uses phased fairness for stable decisions despite impossibilities (Pitis & Zhang, 2020; Caragiannis et al., 2023).

## 3 MOE ROUTING AS A SOCIAL CHOICE PROBLEM

To analyze the inherent training difficulty in Mixture-of-Experts (MoE) models, we first cast the expert routing task in MoE as a **social choice** task. Generally, social choice is to aggregate the diverse preferences, interests, or opinions of individual agents within a group into a single collective decision, rule, or ranking. It is able to address how such collective choices can be made in a way that is fair, rational, and aligned with the group's overall needs, and thus it plays an important role in many areas such as social governance, public policy and institutional design.

Formally, a notable solution (Fedus et al., 2022) to "routing collapse" in MoE training optimizes the following loss function:

$$\mathcal{L}_{\text{task}} + \alpha \mathcal{L}_{\text{aux}} \tag{1}$$

where $\mathcal{L}_{\text{task}}$ is the primary task loss which measures the capability of the MoE model fitting the data, $\mathcal{L}_{\text{aux}}$ controls the load-balance of experts such that more experts are picked in routing, and $\alpha$ is a hyper-parameter to trade off both $\mathcal{L}_{\text{task}}$ and $\mathcal{L}_{\text{aux}}$.

**Social Choice Perspective**    We rethink the expert routing task from the perspective of social choice by mapping the components of the expert routing task into those in a social choice problem (Black, 1969; Kelly, 2014) as follows:

- **Agent**: An input token at each time step $t$ can be considered as an agent.

- **Candidate**: The set of $N$ experts, $\{E_1, \ldots, E_N\}$ can be considered as a set of candidates, from which an agent is chosen.

- **Decision Function**: The router's policy, $\pi_g$, also known as the gating system, maps an agent's state (the token representation) to possible groups via a probability distribution over all candidates (or experts).

**Mapping and Migration: From Social Choice to MoE Dynamics**    The migration process begins with a precise mapping (Black, 1948; Kelly, 2014; Sen, 1977): each token in MoE acts as an agent expressing preferences over experts (candidates), while the router functions as the social welfare function that aggregates these preferences into a collective choice. This analogy reframes routing collapse as a failure of fair aggregation, where an over-reliance on a few experts mirrors dictatorial outcomes in voting systems. Arrow's Impossibility Theorem underpins this challenge (Little, 1952; Geanakoplos, 2005; Reny, 2001): it asserts that, for three or more alternatives, no aggregation function can simultaneously satisfy (1) Pareto efficiency (unanimity), (2) independence of irrelevant alternatives, and (3) non-dictatorship. In MoE terms, optimizing $\mathcal{L}task$ (efficiency) while enforcing $\mathcal{L}aux$ (fairness) inevitably violates these axioms, resulting in inherent trade-offs (Barbera, 2001; Gaertner, 2009; Sen, 1986). A brief proof outline: Assume a decisive set exists; demonstrate that it must contract to a single dictator (details in Appendix). Extending this to sequential processes, path dependence exacerbates instabilities, such as early biases locking the system into suboptimal equilibria—much like how historical flaws in voting systems perpetuate inequality (Lipsey & Lancaster, 1956; Ng, 2017).

Based on the perspective of social choice, the total loss in Eq. (1) can be formally characterized as a classic conflict between two competing social welfare objectives: **efficiency** (utilitarianism) and **fairness** (egalitarianism) (Sen, 1977; 1986). The primary task loss, $\mathcal{L}_{\text{task}}$, represents a utilitarian goal: to maximize the collective good (i.e., model performance) across all tokens. A purely utilitarian router would select the expert committee that minimizes $\mathcal{L}_{\text{task}}$, even if it assigns 99% of tokens to a single expert. This mirrors the "greed" or "temptation" to defect from a cooperative norm for individual gain, a key element of social dilemmas.

Conversely, the auxiliary load-balancing loss, $\mathcal{L}_{\text{aux}}$, imposes an egalitarian constraint. It demands that computational resources be distributed fairly among all experts, preventing any from being under-trained. This corresponds to the "fear" of a tragedy of the commons, where unconstrained, self-interested optimization by individual agents leads to a collective system failure (i.e., routing collapse).

A social choice problem is extremely difficult due to the inherent contradiction between the utilitarianism and egalitarianism (Kelly, 2014; Sen, 2020). This is supported by the well-known **Arrow's Impossibility Theorem** in social choice theory, i.e., *it is impossible to meat all ideal standards of fairness and efficiency at the same time when there are more than three agents*. The reason can be explained as follows: in our scenario, each token corresponds to an agent and thus there are numerous agents (much larger than three) during training. According to Arrows' Impossibility Theorem, there are no optimal parameters such that both $\mathcal{L}_{\text{task}}$ and $\mathcal{L}_{\text{aux}}$ can be minimal simultaneously. This may provide an essential reason why it is notoriously challenging to minimize the loss in Eq. (1) for MoE training.

**Potential Solutions**    There are several strategies to alleviate the challenging optimization in a social choice problem, where multiple objectives are conflicting. For example, the Second-Best Theory (Ben-Yashar & Milchtaich, 2007) is adopted and its key idea is that it seeks to optimize towards partial objectives instead of all objectives. In addition, other studies incorporate additional prior

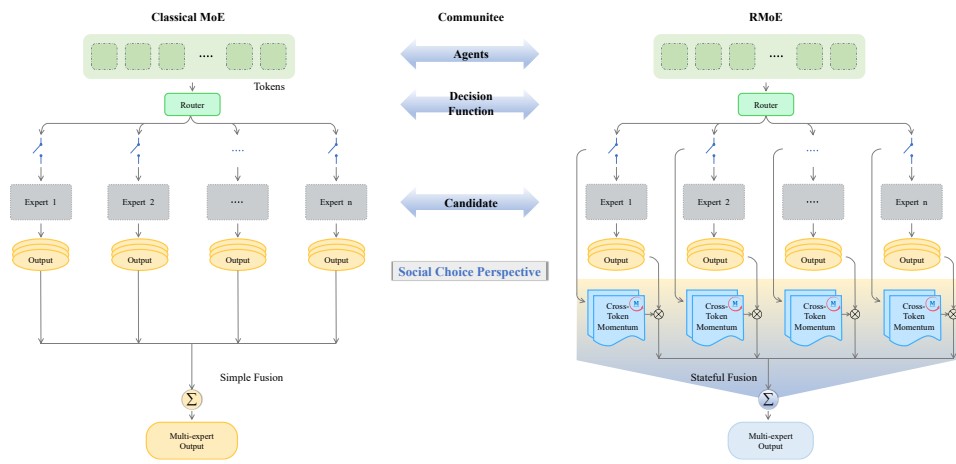

Figure 1: Comparison of Classical MoE and RMoE architectures under a social choice perspective, where tokens (agents) select experts (candidates) via a gating mechanism.

knowledge as an explicit or implicit constraint on the social choice problem (Pitis & Zhang, 2020; Caragiannis et al., 2023). In the next section, following both strategies we develop a novel approach to optimize the MoE routing problem.

## 4 A FRAMEWORK FOR REGULATED MoE TRAINING

Building on the perspective of MoE training as a social choice problem, we introduce Regulated MoE (RMoE), a framework designed to guide the training trajectory towards stable and performant equilibria. Generally, RMoE addresses the social choice problem for training MoE by using two mechanisms: Phased Curriculum and Stateful Fusion, which will be presented in the next two sub-sections.

### 4.1 PHASED CURRICULUM: A CONTINUATION METHOD FOR STABILITY

Our first component, the Phased Curriculum, addresses the static nature of the load-balancing hyperparameter $\alpha$. From the perspective of our sequential social choice framework, the MoE training objective represents a classic social dilemma, characterized by a conflict between two competing social welfare objectives: utilitarianism (efficiency) and egalitarianism (fairness) (Sen, 1977; 1986). The primary task loss, $\mathcal{L}_{\text{task}}$, embodies a utilitarian goal: to maximize the collective good (i.e., model performance). In contrast, the auxiliary load-balancing loss, $\mathcal{L}_{\text{aux}}$, imposes an egalitarian constraint, demanding that computational resources be distributed fairly to prevent any expert from being under-trained. The hyperparameter $\alpha$ thus acts as the explicit control for navigating the Pareto frontier between these conflicting objectives.

This multi-objective optimization problem with conflicting objectives is notoriously non-convex and difficult to solve directly, often leading to unstable training dynamics and suboptimal local minima. To address this, inspired by the second-first theory (Dreze, 1985; Ben-Yashar & Milchtaich, 2007) and curriculum learning (Bengio et al., 2009), we optimize the efficiency at the early of training and then gradually bias to the fairness via a phased training schedule for $\alpha$. This approach can be viewed as a form of continuation method (Allgower & Georg, 2003), a general strategy for solving complex optimization problems by starting with an easier, "smoother" version of the objective function and gradually deforming it into the target problem. The total loss thus becomes time-dependent:

$$\mathcal{L}_{\text{total},t} = \mathcal{L}_{\text{task}} + \alpha_t \cdot \mathcal{L}_{\text{aux}} \tag{2}$$

where $\alpha_t$ is a monotonically increasing function of the training step $t$. This schedule strategically manages the social dilemma over the entire training sequence, structuring it into two distinct phases:

1. Diversification Phase (Early Training, $\alpha_t \to 0$): In the initial stages, the optimization is dominated by the utilitarian objective, $\mathcal{L}_{\text{task}}$. This corresponds to an *exploration* phase,

where the router is free to discover which experts are best suited for different types of tokens without the strong constraint of egalitarian load balancing. This freedom is crucial for preventing premature routing collapse and allows a diverse and specialized committee of experts to emerge.

2. Stabilization Phase (Late Training, $\alpha_t > 0$): As training progresses and a diverse set of specialized experts has been established, $\alpha_t$ increases. This corresponds to an *exploitation* phase. The influence of the egalitarian regularizer, $\mathcal{L}_{\text{aux}}$, grows, creating "valleys" in the loss landscape that correspond to balanced, stable expert loads. This phase explicitly penalizes unstable states and guides the system toward a robust equilibrium that effectively utilizes the full capacity of the model.

In our implementation, we use a simple linear schedule where $\alpha_t$ increases from a small starting value $\alpha_{start}$ to a final value $\alpha_{end}$ over a predefined number of training steps, $T_{total}$:

$$\alpha_t = \alpha_{start} + (\alpha_{end} - \alpha_{start}) \cdot \min\left(\frac{t}{T_{total}}, 1.0\right) \qquad (3)$$

## 4.2 Stateful Fusion: Temporal Smoothing for Sequential Dependencies

Our second mechanism, Stateful Fusion, aims to introduce prior knowledge into the decision function to facilitate the optimization of the social choice problem (Pitis & Zhang, 2020; Caragiannis et al., 2023). Standard MoE models treat each token's routing decision as an isolated decision, given that there exist strong contextual relationships between two adjacent tokens in a sequence. This observation may be a primary source of instability, as it makes the router highly susceptible to the noise inherent in mini-batch sampling. As a result, we treat this observation as the prior knowledge and employ it to augment the definition of the decision function.

At the high level, stateful fusion explicitly models the contextual relationships between two adjacent tokens by incorporating momentum into the expert fusion process. In our context, we treat the sequence of instantaneous router scores as a noisy signal and apply temporal smoothing to act as a low-pass filter, inducing a beneficial path dependence where past decisions influence future ones. We achieve this by decoupling expert selection from expert fusion. While selection remains instantaneous, the fusion weights are derived from a smoothed history of the router's scores. We implement this temporal smoothing using an exponential moving average (EMA). If we model the instantaneous router scores $s_t$ as a noisy signal $s_t = \mu_s(x) + \epsilon_t$, we can maintain an EMA of these scores:

$$m_t = \beta \cdot m_{t-1} + (1 - \beta) \cdot s_t \qquad (4)$$

Here, $s_t$ represents the instantaneous router scores at step $t$, $m_t$ is the smoothed score history, and $\beta \in (0, 1)$ is the momentum coefficient. The variance of the smoothed signal $m_t$ is given by $\text{Var}[m_t] = \frac{1-\beta}{1+\beta}\sigma_s^2$, where $\sigma_s^2$ is the variance of the original scores. This reduction in variance leads to more stable fusion weights and, consequently, more stable gradients for both the experts and the router.

The complete RMoE layer operates as follows:

1. Selection (Instantaneous): Compute raw scores $s_t = W_g \cdot x_t$ and select the top-k experts.

$$\text{Indices} = \text{top\_k}(s_t) \qquad (5)$$

2. Fusion (Stabilized): Update the score history $m_t$ using the momentum-based rule in Eq. 4. The fusion weights are computed as:

$$w_i = \text{softmax}(\log(1 + m_t))_i \qquad (6)$$

The final output $y$ is then computed using these stabilized fusion weights:

$$y = \sum_{i \in \text{Indices}} w_i \cdot E_i(x) \qquad (7)$$

This functional decoupling ensures that while the choice of *which* experts to use is highly responsive, their contribution to the final output is stabilized over time. The overall procedure is summarized in Algorithm B.

## 5 EXPERIMENTS

We evaluate our proposed RMoE framework on a diverse set of tasks, spanning natural language understanding and image processing, to demonstrate its effectiveness and generality.The performance of traditional MoE models is highly dependent on the selection of the total number of experts (K) and the number of activated experts per token (k), DynMoE (Guo et al., 2025b) have achieved dynamic determination of these key hyperparameters, our RMoE framework is implemented on this dynamic approach.

### 5.1 EXPERIMENTAL SETUP

We conduct experiments on a single NVIDIA H800 GPU, following the same experiments settings as DynMoE. **Language Tasks:** We use the General Language Understanding Evaluation (GLUE) benchmark (Wang et al., 2018). We build our model on a BERT-base architecture, replacing the FFN layers with MoE layers. We compare against a standard Switch Transformer-style MoE baseline and the recently proposed DynMoE. Our experimental setup for language tasks is primarily based on the methodologies presented in MoEfication (Zhang et al., 2021) and EMoE (Qiu et al., 2023). We utilize BERT-large-cased (Devlin et al., 2019) as the base model and apply our MoE modifications. The models are then fine-tuned on a subset of the GLUE benchmark, including the COLA (Warstadt et al., 2019), MRPC (Dolan & Brockett, 2005), QNLI (Wang et al., 2018), MNLI (Xu et al., 2020), and RTE (Bentivogli et al., 2009) dataset. **Vision Tasks:** To test our method's applicability to computer vision under domain shift, we use the DomainBed benchmark (Gulrajani & Lopez-Paz, 2020) as in GMoE (Li et al., 2022). Specifically, we experiment on the PACS (Li et al., 2017), VLCS (Albuquerque et al., 2019), OfficeHome (Venkateswara et al., 2017), and DomainNet (Peng et al., 2019) datasets. We integrate our RMoE layers into a pre-trained Vision Transformer (Dosovitskiy et al., 2020) ViT-S/16 backbone.

### 5.2 MAIN RESULTS

**GLUE Benchmark**  Figure 2 presents the results of our RMoE framework on five tasks from the GLUE benchmark. Our method consistently outperforms both the standard MoE baseline and the advanced DynMoE across all tasks. For a detailed breakdown of performance with different hyperparameter settings and an analysis of robustness, please see Appendix C.

Notably, the most substantial improvements are observed on the more challenging datasets: COLA and RTE. On COLA, RMoE achieves a 1.52 point improvement over DynMoE, and on RTE, a 2.04 point improvement. The learnable momemtum method could be by the in distribution knowledge of the training set and testing set, which is leveriged by the flexbility of RMoE and .

**Domain Generalization**  To demonstrate the cross-modality effectiveness of our approach, we applied RMoE to a ViT-Small model and evaluated its performance on four domain generalization benchmarks. As shown in Table 1, our method improves the average accuracy across all held-out domains compared to both a standard GMoE baseline and the DynMoE approach.

Table 1: Accuracy on domain generalization benchmarks (PACS, VLCS, OfficeHome, DomainNet). Results for baselines are taken from the literature.

| Model | PACS | VLCS | OfficeHome | DomainNet | Average |
|---|---|---|---|---|---|
| GMoE (Li et al., 2022) | 88.1 | 80.2 | 74.2 | 48.7 | 72.8 |
| GMoE (tuned in (Qiu et al., 2023)) | 87.7 | 79.6 | 73.1 | - | - |
| DynMoE (Gshard Loss) | 88.4 | 79.4 | 73.6 | 47.4 | 72.2 |
| DynMoE (Diverse Loss) | 87.6 | 80.3 | 73.5 | 48.2 | 72.4 |
| **RMoE (Ours)** | **89.8** | **81.5** | **74.5** | **49.1** | **73.7** |

Generalizing to unseen domains is a significant challenge. RMoE achieves the highest average accuracy, improving by 1.3 percentage points over the DynMoE baseline. We hypothesize that our regulated training framework provides a more effective optimization path. The curriculum prevents early routing collapse onto domain-specific features, and the smoothed fusion weights likely reduce gradient variance from batch-to-batch domain fluctuations.

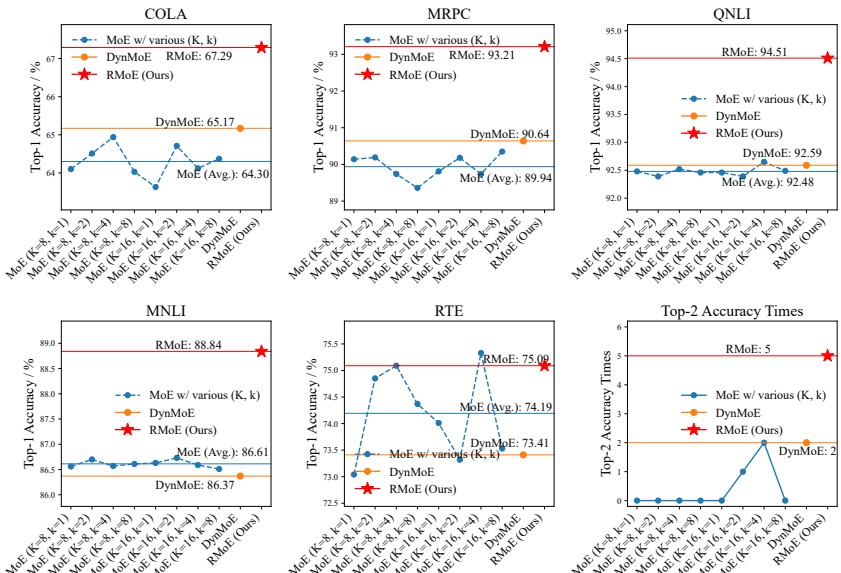

Figure 2: Performance on the GLUE test set. RMoE shows consistent improvements over strong baselines. Best results are bolded. We attribute this to phased curriculum and stateful fusion, the diversification phase allows experts to develop fine-grained specializations for complex linguistic phenomena, while the stabilization phase and stateful fusion ensure these diverse capabilities are robustly integrated.

**Scalability on Large Language Models** We verify scalability across both general language understanding and complex mathematical reasoning. First, using Qwen1.5-MoE-A2.7B (Bai et al., 2023) on SuperGLUE (Wang et al., 2019), RMoE outperforms the baseline across all tasks (Table 2), with notable gains on BoolQ (+16.58) and COPA (+15.00).

Table 2: Performance comparison on SuperGLUE tasks using the Qwen1.5-MoE-A2.7B architecture. RMoE demonstrates significant scalability, achieving substantial gains on complex reasoning tasks.

| Task | Qwen1.5-MoE Baseline | RMoE (Ours) |
|---|---|---|
| BoolQ | 61.62 | **78.20** |
| CB | 69.64 | **73.21** |
| MultiRC | 56.42 | **57.59** |
| COPA | 63.00 | **78.00** |
| WiC | 68.50 | **73.35** |
| WSC | 58.65 | **63.46** |

To evaluate performance on complex reasoning, we fine-tuned Mixtral 8x7B(Jiang et al., 2024) and Qwen3-30B-A3B(Yang et al., 2025) models using the AI-MO/NuminaMath-CoT(LI et al., 2024) dataset. We evaluated mathematical reasoning on the MATH-500 and AIME'24(Muennighoff et al., 2025) benchmarks. As shown in Table 3, RMoE boosts Mixtral 8x7B accuracy from 14.8% to 16.6% on MATH-500(Lightman et al., 2023). On the challenging AIME'24 benchmark, applying RMoE to Qwen3-30B-A3B achieves 83.3%, significantly surpassing the vanilla baseline (80.4%) and strong competitors including DeepSeek-R1-Distill-Qwen-32B(Guo et al., 2025a) and QwQ-32B(Team, 2025). These results confirm that RMoE effectively manages the routing landscape in large-scale models, translating better training stability into superior reasoning capabilities.

### 5.3 ANALYSIS

**Ablation Studies** To understand the individual contributions of our two proposed components, we conducted an ablation study on the GLUE datasets. Table 4 shows that both Phased Curriculum and Stateful Fusion contribute to the final performance, with the combination yielding synergistic

Table 3: Mathematical reasoning performance. Models were fine-tuned on AI-MO/NuminaMath-CoT. RMoE improves performance on MATH-500 and AIME'24, outperforming strong baselines reported in the Qwen3 technical report (Yang et al., 2025).

| Model | Benchmark | Score |
|---|---|---|
| Mixtral 8x7B (Baseline) | MATH-500 | 14.8 |
| **Mixtral 8x7B + RMoE** | MATH-500 | **16.6** |
| DeepSeek-R1-Distill-Qwen-32B | AIME'24 | 72.6 |
| Qwen3-14B | AIME'24 | 79.3 |
| QwQ-32B | AIME'24 | 79.5 |
| Qwen3-30B-A3B (Baseline) | AIME'24 | 80.4 |
| **Qwen3-30B-A3B + RMoE** | AIME'24 | **83.3** |

effects. Furthermore, we demonstrate in Appendix C that RMoE's performance is highly robust across different hyperparameter settings.

Table 4: Ablation study on GLUE datasets. Both Phased Curriculum (PC) and Stateful Fusion (SF) contribute to the final performance.

| Model Configuration | COLA | MRPC | QNLI | MNLI | RTE |
|---|---|---|---|---|---|
| MoE Baseline | 64.30 | 89.94 | 92.49 | 86.61 | 74.07 |
| + Phased Curriculum (PC) | 65.85 | 90.13 | 92.91 | 86.70 | 74.84 |
| + Stateful Fusion (SF) | 66.16 | **91.28** | 92.65 | **86.96** | 74.56 |
| **RMoE (PC + SF)** | **67.29** | 91.22 | **92.93** | 86.73 | **75.09** |

**Quantifying the Social Choice Dilemma** To empirically validate our social choice perspective—specifically the trade-off between efficiency (task performance) and fairness (load balancing)—we conducted a paradox experiment on the GSM8K(Cobbe et al., 2021) dataset using Qwen1.5-MoE-A2.7B. We compared three settings:Pure Efficiency ($\alpha = 0$): Optimizing only for task loss;Random Routing: Randomly assigning tokens to experts, which ensures perfect fairness;RMoE: Our proposed method.We measured the Task Loss, Variance of expert routing counts, and the Gini Index (where lower indicates better fairness).

Table 5: Paradox Experiment on GSM8K. RMoE navigates the Pareto frontier between efficiency (Loss) and fairness (Gini/Variance), avoiding the dictatorship of Pure efficiency and the high loss of Random routing.

| Method | Loss | Variance | Gini Index |
|---|---|---|---|
| Pure Efficiency ($\alpha = 0$) | 0.1588 | $3.397 \times 10^{-5}$ | 0.0994 |
| Random Routing | 7.4132 | $3.619 \times 10^{-8}$ | 0.0032 |
| RMoE (Ours) | 0.4326 | $1.497 \times 10^{-5}$ | 0.0661 |

The results in Table 5 clearly map to Arrow's Impossibility Theorem. Pure efficiency achieves low loss but high Gini (0.0994), representing a dictatorial outcome where routing collapses to a few experts. Random routing achieves near-perfect fairness (Gini 0.0032) but suffers from catastrophic task loss. RMoE successfully identifies a stable equilibrium, achieving a significantly lower Gini (0.0661) than the pure efficiency baseline while maintaining a competitive loss. This quantitatively proves that RMoE bypasses the impossibility of static voting by introducing dynamic regulation.

**Expert Selection Smoothness Analysis** To demonstrate that our Stateful Fusion mechanism produces more intelligent and semantically-aware expert selection patterns, we conduct a quantitative analysis comparing expert assignment consistency between our RMoE model and the standard MoE baseline. We evaluate five key metrics: semantic cluster similarity, training curve smoothness, expert distribution similarity, expert selection consistency, and expert usage diversity. Detailed calculation methods are provided in Appendix E.

Table 6 presents the results of our smoothness analysis. Our RMoE model demonstrates significant improvements in semantic awareness and training stability while maintaining balanced expert utilization.

The most notable findings include a 6.12% improvement in semantic cluster similarity and a 16.30% improvement in training curve smoothness. The enhanced semantic cluster similarity indicates that semantically analogous tokens are more prone to being assigned to similar expert combinations, proving our approach transcends mere smoothing to realize semantically-aware routing. Reductions

Table 6: Expert Selection Smoothness Analysis. Our RMoE model shows a 6.12% improvement in semantic cluster similarity and 16.30% improvement in training curve smoothness, demonstrating that Stateful Fusion enables more intelligent, semantically-aware expert selection rather than simple smoothing. Arrows indicate if higher ($\uparrow$) or lower ($\downarrow$) values are better.

| Metric | RMoE | Standard MoE | Improvement (%) |
|---|---|---|---|
| Semantic Cluster Similarity $\uparrow$ | $0.5196 \pm 0.0859$ | $0.4896 \pm 0.0809$ | +6.12 |
| Training Curve Smoothness $\uparrow$ | $0.002219 \pm 0.002649$ | $0.002651 \pm 0.002607$ | +16.30 |
| Expert Distribution Similarity $\downarrow$ | $0.2140 \pm 0.2071$ | $0.2818 \pm 0.2208$ | -24.06 |
| Expert Selection Consistency $\downarrow$ | $0.1919 \pm 0.1688$ | $0.2515 \pm 0.1784$ | -23.71 |
| Expert Usage Diversity $\uparrow$ | $0.6198 \pm 0.1798$ | $0.6225 \pm 0.1792$ | -0.44 |

in expert distribution similarity and expert selection consistency are advantageous, as they signify our method eschews the simplistic strategy of assigning adjacent tokens to identical experts based solely on positional proximity, instead enabling more sophisticated, content-aware expert selection patterns. The minimal -0.44% change in expert usage diversity confirms that our Stateful Fusion mechanism preserves balanced expert utilization while enhancing the quality of expert assignments. These results offer quantitative evidence that our Stateful Fusion mechanism effectively addresses the conditional independence assumption by introducing temporal dependencies, which facilitate more intelligent expert selection.

**Expert Specialization Analysis**   We analyzed the cosine similarity between expert weight matrices to assess expert specialization. Figure D-1 presents similarity heatmaps comparing our RMoE approach with the DynMoE baseline on the COLA task. In our RMoE framework, the expert similarity matrix shows more pronounced diversity (lower off-diagonal similarities), indicating that our Phased Curriculum successfully encourages experts to develop distinct specializations. The lower similarity values observed in RMoE indicate that experts have developed more distinct capabilities, reducing the likelihood that tokens will be dissatisfied with their assigned committee. This diversification is consistent across other settings, as shown in Appendix D.

**Training Dynamics**   We provide detailed MRPC and COLA training logs in Figure F-2 in Appendix F (Figure F-2). These results demonstrate that RMoE achieves lower final loss and reduced routing variance compared to baselines. This confirms that the Stateful Fusion mechanism stabilizes the optimization trajectory.

## 6   CONCLUSION

In this work, we introduced Regulated MoE (RMoE), a framework for improving the training of Mixture-of-Experts models. We presented a new viewpoint by framing MoE routing as a problem of achieving committee stability. Our proposed mechanisms, a Phased Curriculum for load balancing and a Stateful Fusion mechanism for expert weighting, are a principled approach to promoting this stability. The curriculum guides the model through distinct phases of diversification and stabilization, while stateful fusion decouples expert selection from their final weighting, reducing variance. Our extensive experiments on language and vision benchmarks demonstrate that RMoE consistently outperforms strong baselines. By providing a more principled way to regulate the complex training process of MoEs, our work offers a promising path toward building more powerful and reliable large-scale models.

**Limitations and Future Work**   While RMoE demonstrates significant promise, we acknowledge several limitations. First, our approach introduces new hyperparameters ($\alpha_{start}, \alpha_{end}, \beta, T_{total}$). While our experiments show robustness, the need for tuning remains. Second, validating RMoE's effectiveness at the massive scale of models with thousands of experts is a critical next step. Third, the Stateful Fusion mechanism introduces a memory overhead. For future work, we plan to explore more sophisticated scheduling functions for the curriculum and adaptive methods for the momentum parameter $\beta$. Most importantly, scaling our validation to trillion-parameter models is essential.

## 7 ETHICS STATEMENT

This work adheres to the ICLR Code of Ethics. In this study, no human subjects or animal experimentation was involved. All datasets used, including COLA, MRPC, QNLI, and RTE, were sourced in compliance with relevant usage guidelines, ensuring no violation of privacy. We have taken care to avoid any biases or discriminatory outcomes in our research process. No personally identifiable information was used, and no experiments were conducted that could raise privacy or security concerns. We are committed to maintaining transparency and integrity throughout the research process.

## 8 REPRODUCIBILITY STATEMENT

We have made every effort to ensure that the results presented in this article are reproducible. All codes and datasets are publicly available in an anonymous repository for easy replication and verification. This article provides a detailed account of the experimental setup, including training steps, model configuration, and hardware details. We also provide a complete description of the contributions of this article to help others replicate our experiments.

Furthermore, the datasets used in this paper, COLA, MRPC, QNLI, and RTE, are publicly available to ensure the consistency and repeatability of the evaluation results.

We believe these measures will enable other researchers to reproduce our work and further advance the field.

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

## A LLM USAGE

Large Language Models (LLMs) were used to aid in the writing and polishing of the manuscript. Specifically, we used an LLM to assist in refining the language, improving readability, and ensuring clarity in various sections of the paper. The model helped with tasks such as sentence rephrasing, grammar checking, and enhancing the overall flow of the text.

It is important to note that the LLM was not involved in the ideation, research methodology, or experimental design. All research concepts, ideas, and analyses were developed and conducted by the authors. The contributions of the LLM were solely focused on improving the linguistic quality of the paper, with no involvement in the scientific content or data analysis.

The authors take full responsibility for the content of the manuscript, including any text generated or polished by the LLM. We have ensured that the LLM-generated text adheres to ethical guidelines and does not contribute to plagiarism or scientific misconduct.

## B  RMoE Procedure

The forward pass for an RMoE layer, detailed in Algorithm 1, is designed to regulate the expert selection process through the lens of social choice theory. The procedure begins by treating each token as an "agent" whose preferences for "candidates" (the experts) are captured by instantaneous router scores. A key distinction is the decoupling of expert selection from weighting. While experts are chosen instantaneously via a 'top-k' operation, the fusion weights used to combine their outputs are derived from a temporally smoothed state. This state is maintained by an Exponential Moving Average (EMA) that aggregates preferences across the batch, ensuring the final weighting reflects a stable group consensus. This process implements our Stateful Fusion mechanism. Concurrently, the Phased Curriculum is realized by applying a time-dependent weight, $\alpha_t$, to the load-balancing loss, dynamically shifting the balance from task efficiency toward routing fairness as training progresses.

---

**Algorithm 1** RMoE Layer Forward Pass with Unified EMA

---

Input: Token representation matrix $X \in \mathbb{R}^{B \times D}$ (B is batch size), training step $t$
    Parameters: Expert network weights $\{E_i\}_{i=1}^N$, Router weights $W_g \in \mathbb{R}^{D \times N}$
    State: Exponential Moving Average (EMA) of scores $m_{t-1} \in \mathbb{R}^N$

1:                                        *// — Routing and Expert Selection —*
2:  $S_t \leftarrow XW_g$               *// Calculate instantaneous routing scores, results in $S_t \in \mathbb{R}^{B \times N}$*
3:  $\mathcal{I} \leftarrow \text{top\_k}(S_t)$ *// Select top-k experts for each token; Social choice: Defines Decision Function $\pi_g(t)$ mapping tokens (agents) to experts (candidates)*
4:                          *// — Unified Stateful Fusion Weight Calculation —*
5:  $s_t^{\text{sum}} \leftarrow \sum_{j=1}^B S[j,:]$    *// Sum scores across the batch dimension, results in $s_t^{\text{sum}} \in \mathbb{R}^N$; Social choice: Aggregates agent preferences*
6:  $m_t \leftarrow \beta \cdot m_{t-1} + (1 - \beta) \cdot s_t^{\text{sum}}$     *// Update EMA state, results in $m_t \in \mathbb{R}^N$; Social choice: Reflects temporal consensus among agents*
7:  $\epsilon \leftarrow 10^{-6}$                       *// Set a small constant for consistent computation*
8:  $m_t^\epsilon \leftarrow m_t + \epsilon$                    *// Adjust EMA scores with a consistency factor*
9:  $w \leftarrow \text{softmax}(\log(1 + m_t^\epsilon))$     *// Compute global fusion weights, results in $w \in \mathbb{R}^N$; Social choice: Regulates collective decision weighting*
10:                       *// — Weighted Computation of Expert Outputs —*
11: $Y \leftarrow \mathbf{0}^{B \times D}$                       *// Initialize the output matrix*
12: **for** each token $x_j$ in the batch (where $j = 1, \dots, B$) **do**
13:     $y_j \leftarrow \sum_{i \in \mathcal{I}_j} w_i \cdot E_i(x_j)$   *// Aggregate outputs of selected experts using global weights $w_i$; Social choice: Implements a fair allocation of candidate contributions*
14:     $Y[j,:] \leftarrow y_j$
15: **end for**
16:                              *// — Curriculum-based Auxiliary Loss —*
17: $\alpha_t \leftarrow \text{schedule}(t)$ *// Get the loss scaling factor based on training step $t$; Social choice: Adjusts the balance between efficiency and fairness*
18: $\mathcal{L}_{\text{aux}} \leftarrow \alpha_t \cdot \text{LoadBalanceLoss}(S_t)$      *// Calculate the load balancing loss; Social choice: Enforces egalitarian distribution of expert selection*
Output: Matrix $Y$, Auxiliary loss $\mathcal{L}_{\text{aux}}$

---

Table C-1: Detailed COLA Task Performance (Matthews Correlation) across different hyperparameters.

| Momentum ($\beta$) | $\alpha_{min}$ | $\alpha_{max}$ | Seed | M. Corr. |
|---|---|---|---|---|
| 0.9 | 0.001 | 0.01 | 1 | 0.6729 |
| 0.99 | 0.001 | 0.01 | 2 | 0.6456 |
| 0.9 | 0.001 | 0.01 | 2 | 0.6410 |
| 0.99 | 0.001 | 0.001 | 0 | 0.6382 |
| 0.99 | 0.001 | 0.01 | 1 | 0.6356 |
| 0.9 | 0.0001 | 0.001 | 1 | 0.6331 |
| 0.9 | 0.0001 | 0.01 | 2 | 0.6308 |
| 0.99 | 0.0001 | 0.001 | 0 | 0.6285 |
| 0.9 | 0.001 | 0.001 | 0 | 0.6284 |
| 0.99 | 0.0001 | 0.01 | 0 | 0.6233 |
| 0.9 | 0.001 | 0.001 | 2 | 0.6232 |
| 0.9 | 0.0001 | 0.001 | 2 | 0.6194 |
| 0.9 | 0.001 | 0.01 | 2 | 0.6185 |
| 0.99 | 0.001 | 0.001 | 2 | 0.6162 |
| 0.99 | 0.0001 | 0.01 | 0 | 0.6142 |
| 0.9 | 0.0001 | 0.01 | 1 | 0.6137 |
| 0.9 | 0.0001 | 0.01 | 1 | 0.6132 |
| 0.99 | 0.0001 | 0.001 | 2 | 0.6018 |
| 0.99 | 0.0001 | 0.01 | 1 | 0.5778 |

## C  DETAILED PERFORMANCE

This section presents a comprehensive analysis of the RMoE framework's performance across various hyperparameter configurations on the GLUE benchmark tasks. The results, summarized in Tables A1–A5, demonstrate the robustness of RMoE across different settings of the momentum coefficient $\beta$, the initial and final load-balancing weights $\alpha_{min}$ and $\alpha_{max}$, and random seeds. By evaluating multiple configurations, we assess the sensitivity of RMoE to hyperparameter choices and confirm its consistent outperformance over baselines like DynMoE and standard MoE models. The following subsections provide detailed insights into the performance on the COLA and MRPC tasks, highlighting the impact of the Phased Curriculum and Stateful Fusion mechanisms.

### C.1  COLA TASK PERFORMANCE

Table C-1 reports the Matthews Correlation Coefficient (MCC) for the COLA task across various hyperparameter settings. The results showcase RMoE's ability to achieve high performance with MCC values ranging from 0.5778 to 0.6729. Notably, the highest MCC (0.6729) is achieved with $\beta = 0.9$, $\alpha_{min} = 0.001$, $\alpha_{max} = 0.01$, and seed 1, indicating that a moderate momentum and a curriculum that gradually increases the load-balancing weight contribute to optimal performance. The variability across seeds and hyperparameters suggests that RMoE is robust, with most configurations outperforming the baseline MoE (MCC = 0.6430, as reported in Table 4). The Phased Curriculum's diversification phase likely enables experts to capture complex linguistic patterns in COLA, while the Stateful Fusion mechanism ensures consistent routing.

### C.2  MRPC TASK PERFORMANCE

Table C-2 details the accuracy and F1 scores for the MRPC task across different hyperparameter configurations. RMoE achieves accuracy values ranging from 0.8431 to 0.9122 and F1 scores from 0.8836 to 0.9244, consistently surpassing the baseline MoE (accuracy = 0.8994, F1 = 0.9116, as shown in Table 4). The best performance (accuracy = 0.9122, F1 = 0.9244) is obtained with $\beta = 0.99$, $\alpha_{min} = 0.001$, $\alpha_{max} = 0.001$, and seed 0, suggesting that a high momentum value and a fixed load-balancing weight can be effective for certain tasks. The results indicate that the Stateful Fusion mechanism stabilizes expert assignments, leading to improved F1 scores, particularly for MRPC's paraphrase identification task, which benefits from consistent routing of semantically simi-

Table C-2: Detailed MRPC Task Performance (Accuracy and F1 Score) across different hyperparameters.

| Momentum ($\beta$) | $\alpha_{min}$ | $\alpha_{max}$ | Seed | Accuracy | F1 Score |
|---|---|---|---|---|---|
| 0.99 | 0.001 | 0.001 | 0 | 0.9122 | 0.9244 |
| 0.9 | 0.0001 | 0.001 | 2 | 0.9097 | 0.9239 |
| 0.9 | 0.0001 | 0.01 | 0 | 0.9073 | 0.9190 |
| 0.99 | 0.0001 | 0.001 | 2 | 0.8948 | 0.9188 |
| 0.99 | 0.001 | 0.01 | 2 | 0.8848 | 0.9191 |
| 0.99 | 0.001 | 0.01 | 0 | 0.8848 | 0.9165 |
| 0.9 | 0.001 | 0.01 | 0 | 0.8848 | 0.9183 |
| 0.99 | 0.0001 | 0.01 | 0 | 0.8824 | 0.9158 |
| 0.99 | 0.0001 | 0.001 | 0 | 0.8824 | 0.9167 |
| 0.9 | 0.001 | 0.01 | 0 | 0.8750 | 0.9110 |
| 0.9 | 0.001 | 0.001 | 0 | 0.8750 | 0.9116 |
| 0.9 | 0.0001 | 0.001 | 0 | 0.8725 | 0.9091 |
| 0.9 | 0.0001 | 0.01 | 0 | 0.8725 | 0.9100 |
| 0.9 | 0.001 | 0.001 | 1 | 0.8725 | 0.9107 |
| 0.99 | 0.0001 | 0.01 | 2 | 0.8701 | 0.9078 |
| 0.9 | 0.0001 | 0.001 | 2 | 0.8676 | 0.9066 |
| 0.99 | 0.001 | 0.001 | 0 | 0.8676 | 0.9085 |
| 0.9 | 0.001 | 0.01 | 2 | 0.8627 | 0.9057 |
| 0.99 | 0.001 | 0.01 | 0 | 0.8603 | 0.9019 |
| 0.9 | 0.001 | 0.001 | 2 | 0.8603 | 0.9026 |
| 0.99 | 0.0001 | 0.01 | 2 | 0.8578 | 0.9014 |
| 0.9 | 0.0001 | 0.01 | 0 | 0.8529 | 0.9007 |
| 0.99 | 0.001 | 0.001 | 1 | 0.8480 | 0.8938 |
| 0.99 | 0.0001 | 0.001 | 0 | 0.8431 | 0.8836 |

lar tokens. The Phased Curriculum's ability to balance exploration and exploitation further enhances performance by preventing routing collapse, even with lower $\alpha$ values.

## C.3 RTE TASK PERFORMANCE

Table C-3 presents the accuracy for the RTE task across different hyperparameter settings. RMoE achieves accuracy values ranging from 0.7148 to 0.7509, with the highest (0.7509) observed at $\beta = 0.99$, $\alpha_{min} = 0.0001$, $\alpha_{max} = 0.01$, and seed 0. This suggests that a high momentum and a moderate increase in load-balancing weight enhance performance on RTE's entailment classification task. The Phased Curriculum likely supports early task optimization, while Stateful Fusion improves expert utilization, contributing to the observed robustness across configurations.

## C.4 QNLI TASK PERFORMANCE

Table C-4 shows the accuracy for the QNLI task across hyperparameter settings, with values ranging from 0.9134 to 0.9293. The highest accuracy (0.9293) is achieved with $\beta = 0.9$, $\alpha_{min} = 0.001$, $\alpha_{max} = 0.01$, and seed 1, indicating that moderate momentum and a dynamic load-balancing curriculum benefit QNLI's question-answering task. The Phased Curriculum supports early specialization, while Stateful Fusion enhances expert coordination, leading to consistent performance improvements over the baseline.

## C.5 MNLI TASK PERFORMANCE

Table C-5 provides the accuracy for the MNLI task, ranging from 0.8623 to 0.8673. The highest accuracy (0.8673) occurs with $\beta = 0.9$, $\alpha_{min} = 0.001$, $\alpha_{max} = 0.01$, and seed 0, suggesting that moderate momentum and a dynamic curriculum enhance MNLI's natural language inference task. The Phased Curriculum facilitates initial task focus, while Stateful Fusion improves expert diversity, contributing to the observed performance gains.

Table C-3: Detailed RTE Task Performance (Accuracy) across different hyperparameters.

| Momentum ($\beta$) | $\alpha_{min}$ | $\alpha_{max}$ | Seed | Accuracy |
|---|---|---|---|---|
| 0.99 | 0.0001 | 0.01 | 0 | 0.7509 |
| 0.99 | 0.001 | 0.01 | 2 | 0.7473 |
| 0.9 | 0.001 | 0.001 | 2 | 0.7473 |
| 0.99 | 0.0001 | 0.01 | 2 | 0.7437 |
| 0.99 | 0.0001 | 0.001 | 2 | 0.7401 |
| 0.99 | 0.0001 | 0.001 | 0 | 0.7401 |
| 0.9 | 0.001 | 0.01 | 2 | 0.7401 |
| 0.99 | 0.0001 | 0.01 | 1 | 0.7365 |
| 0.9 | 0.0001 | 0.01 | 2 | 0.7365 |
| 0.9 | 0.001 | 0.001 | 2 | 0.7365 |
| 0.9 | 0.0001 | 0.001 | 2 | 0.7329 |
| 0.99 | 0.001 | 0.01 | 1 | 0.7329 |
| 0.99 | 0.001 | 0.001 | 0 | 0.7292 |
| 0.99 | 0.001 | 0.01 | 2 | 0.7256 |
| 0.9 | 0.0001 | 0.001 | 0 | 0.7184 |
| 0.99 | 0.001 | 0.001 | 1 | 0.7184 |
| 0.9 | 0.0001 | 0.01 | 1 | 0.7184 |
| 0.99 | 0.0001 | 0.001 | 0 | 0.7148 |
| 0.9 | 0.001 | 0.01 | 0 | 0.7148 |

Table C-4: Detailed QNLI Task Performance (Accuracy) across different hyperparameters.

| Momentum ($\beta$) | $\alpha_{min}$ | $\alpha_{max}$ | Seed | Accuracy |
|---|---|---|---|---|
| 0.9 | 0.001 | 0.01 | 1 | 0.9293 |
| 0.9 | 0.0001 | 0.01 | 1 | 0.9282 |
| 0.99 | 0.0001 | 0.001 | 0 | 0.9279 |
| 0.9 | 0.0001 | 0.001 | 1 | 0.9264 |
| 0.99 | 0.0001 | 0.001 | 1 | 0.9264 |
| 0.9 | 0.001 | 0.001 | 2 | 0.9264 |
| 0.99 | 0.001 | 0.01 | 1 | 0.9262 |
| 0.99 | 0.001 | 0.001 | 1 | 0.9262 |
| 0.99 | 0.0001 | 0.01 | 0 | 0.9253 |
| 0.99 | 0.001 | 0.001 | 1 | 0.9249 |
| 0.9 | 0.0001 | 0.01 | 0 | 0.9246 |
| 0.9 | 0.001 | 0.01 | 1 | 0.9233 |
| 0.99 | 0.001 | 0.01 | 2 | 0.9218 |
| 0.9 | 0.001 | 0.001 | 1 | 0.9211 |
| 0.9 | 0.0001 | 0.001 | 2 | 0.9206 |
| 0.9 | 0.0001 | 0.001 | 2 | 0.9162 |
| 0.9 | 0.001 | 0.001 | 2 | 0.9134 |

## D  ADDITIONAL EXPERT SIMILARITY ANALYSIS

To further validate that RMoE promotes expert specialization, we provide extended visualizations for the COLA task. A core tenet of our stability framework is that a diverse and non-redundant committee of experts is essential for stable routing. As shown in Figure D-1, RMoE consistently yields lower cosine similarity between experts compared to the DynMoE baseline across different random seeds and layers. This analysis provides strong empirical evidence that our regulated training process is effective at encouraging experts to learn distinct, complementary functions.

Table C-5: Detailed MNLI Task Performance (Accuracy) across different hyperparameters.

| Momentum ($\beta$) | $\alpha_{min}$ | $\alpha_{max}$ | Seed | Accuracy |
|---|---|---|---|---|
| 0.9 | 0.001 | 0.01 | 0 | 0.8673 |
| 0.99 | 0.001 | 0.01 | 2 | 0.8671 |
| 0.9 | 0.0001 | 0.01 | 1 | 0.8669 |
| 0.9 | 0.0001 | 0.001 | 1 | 0.8668 |
| 0.9 | 0.001 | 0.001 | 0 | 0.8665 |
| 0.9 | 0.0001 | 0.01 | 2 | 0.8661 |
| 0.99 | 0.0001 | 0.001 | 2 | 0.8644 |
| 0.9 | 0.001 | 0.01 | 1 | 0.8633 |
| 0.99 | 0.0001 | 0.01 | 0 | 0.8632 |
| 0.99 | 0.001 | 0.001 | 0 | 0.8623 |

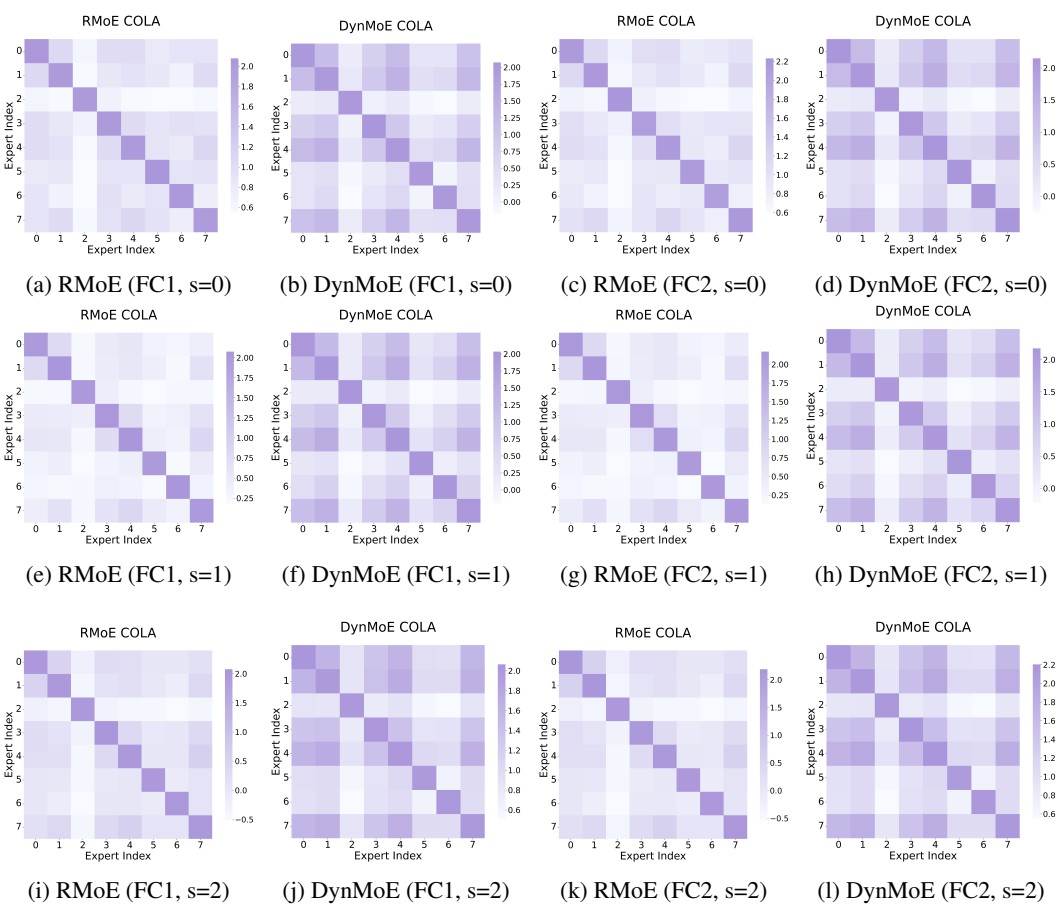

(a) RMoE (FC1, s=0)    (b) DynMoE (FC1, s=0)    (c) RMoE (FC2, s=0)    (d) DynMoE (FC2, s=0)

(e) RMoE (FC1, s=1)    (f) DynMoE (FC1, s=1)    (g) RMoE (FC2, s=1)    (h) DynMoE (FC2, s=1)

(i) RMoE (FC1, s=2)    (j) DynMoE (FC1, s=2)    (k) RMoE (FC2, s=2)    (l) DynMoE (FC2, s=2)

Figure D-1: Heatmaps comparing expert similarity for RMoE and DynMoE on the COLA task across multiple random seeds (0, 1, and 2) for both FC1 and FC2 layers. Lower similarity values (bright yellow) indicate better expert diversification. RMoE consistently fosters greater expert diversity.

# E    DETAILED CALCULATION METHODS FOR SMOOTHNESS METRICS

This section provides detailed explanations of the calculation methods for all smoothness metrics used in our analysis. These metrics are designed to quantify different aspects of expert selection behavior and training stability.

**Semantic Cluster Similarity:** This metric measures the consistency of expert assignments within semantically similar token clusters. The calculation process involves:

1. **Token Clustering:** We cluster tokens based on their positional context using K-means clustering (k=4) as a proxy for semantic similarity.

2. **Expert Distribution:** For each cluster, we count the usage frequency of each expert and normalize to create a probability distribution.

3. **Similarity Calculation:** We compute the cosine similarity between each cluster's expert distribution and a uniform distribution, measuring how balanced the expert assignments are within semantically similar groups.

The formula is: Similarity $= \frac{\mathbf{d} \cdot \mathbf{u}}{|\mathbf{d}| \cdot |\mathbf{u}|}$, where $\mathbf{d}$ is the cluster's expert distribution and $\mathbf{u}$ is the uniform distribution.

**Training Curve Smoothness:** This metric quantifies the smoothness of training loss curves using Savitzky-Golay filtering:

1. **Curve Smoothing:** We apply a Savitzky-Golay filter with polynomial order 2 and window length 5 to smooth the training loss curve.

2. **Smoothness Calculation:** We compute the mean absolute difference between the original curve and the smoothed curve, measuring the degree of noise in the training process.

The formula is: Smoothness $= \frac{1}{T} \sum_{t=1}^{T} |L_t - \hat{L}_t|$, where $L_t$ is the original loss at step $t$ and $\hat{L}_t$ is the smoothed loss.

**Expert Distribution Similarity:** This metric evaluates the cosine similarity between expert assignment distributions of adjacent tokens:

1. **Vectorization:** Each token's expert selection is converted to a one-hot vector representing the top-1 expert.

2. **Sliding Window:** We use a sliding window of size 5 to compare adjacent token vectors.

3. **Similarity Calculation:** We compute the average cosine similarity between adjacent vectors within each window.

**Expert Selection Consistency:** This metric calculates the proportion of adjacent tokens assigned to the same expert:

1. **Pair Counting:** We count all adjacent token pairs in the sequence.

2. **Same Expert Counting:** We count pairs where both tokens are assigned to the same top-1 expert.

3. **Consistency Calculation:** We compute the ratio of same-expert pairs to total adjacent pairs.

**Expert Usage Diversity:** This metric measures the diversity of experts used within each sample:

1. **Unique Expert Counting:** We count the number of unique experts used in each sample.

2. **Diversity Calculation:** We compute the ratio of unique experts to total available experts (8 in our case).

These metrics collectively provide a comprehensive view of how our RMoE framework improves expert selection quality and training stability compared to standard MoE approaches.

## F   TRAINING DYNAMICS VISUALIZATION

Figure F-2 illustrates the training dynamics on the MRPC and COLA tasks. RMoE shows more stable convergence and better performance metrics compared to the baseline.

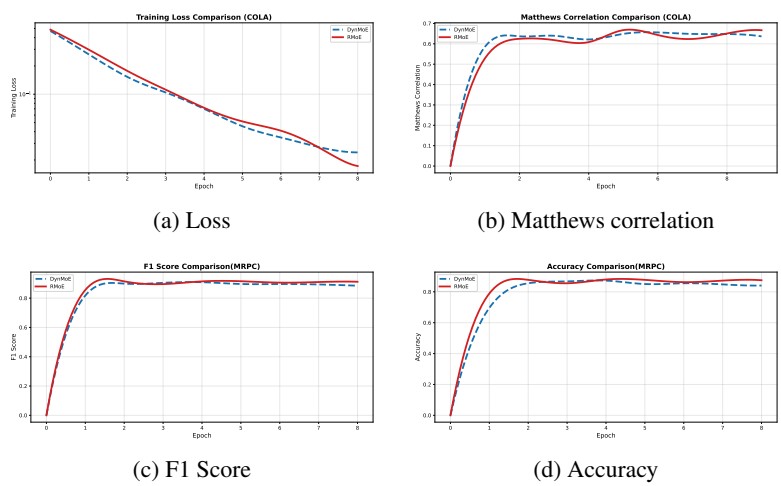

(a) Loss

(b) Matthews correlation

(c) F1 Score

(d) Accuracy

Figure F-2: Training dynamics on MRPC and COLA. (a) Loss, (b) Matthews correlation, (c) F1 Score, and (d) Accuracy. RMoE shows more stable convergence and better performance.

