# OpenReview forum: "On Training Mixture-of-Experts: A Social Choice Perspective"
_ICLR.cc/2026/Conference — Submitted to ICLR 2026_

### Official Review · Reviewer_vmUk · 2025-10-29

**Soundness:** 1
**Presentation:** 1
**Contribution:** 2
**Rating:** 2
**Confidence:** 3

**Summary:**

This paper reframes MoE training through the lens of social choice theory, arguing that the difficulty in balancing task performance and load balancing can be attributed to Arrow's Impossibility Theorem. The authors propose RMoE, which combines a phased curriculum for the load-balancing loss weight and stateful fusion using momentum for expert weighting. They demonstrate improvements over baselines on GLUE and domain generalization benchmarks.

**Strengths:**

1. The paper attempts an interesting interdisciplinary connection between MoE training and social choice theory, which offers new insights into understanding routing collapse.

2. The experimental results show consistent improvements over baselines across multiple benchmarks.

3.  the paper includes extensive ablation studies and analysis of expert specialization patterns.

**Weaknesses:**

1. The social choice framing feels more like a loose analogy than a rigorous theoretical foundation. While the paper invoke Arrow's Impossibility Theorem, they don't provide a formal proof of how it applies to MoE training - the mapping between voting systems and routing is imprecise. The actual solutions proposed  are fairly standard techniques that don't really emerge from social choice principles. For instance, curriculum-based approaches for MoE have been explored in [1] and progressive training strategies in [2].

2. The technical contributions are incremental. The phased curriculum is essentially scheduling the auxiliary loss weight, which has been explored in various forms (e.g., gradual unfreezing, warm-up schedules). The stateful fusion mechanism is just applying EMA to routing scores, similar to momentum-based methods in optimization. Neither innovation requires the social choice perspective to motivate or understand.

3. The experimental setup has significant limitations. All experiments use relatively small models, making it unclear if the approach scales to production-scale MoE models like Mixtral-8x7B [3] where routing collapse is more problematic.

[1] Lewis M, Bhosale S, Dettmers T, et al. Base layers: Simplifying training of large, sparse models[C]//International Conference on Machine Learning. PMLR, 2021: 6265-6274.
[2] Zhou Y, Lei T, Liu H, et al. Mixture-of-experts with expert choice routing[J]. Advances in Neural Information Processing Systems, 2022, 35: 7103-7114.
[3] Jiang A Q, Sablayrolles A, Roux A, et al. Mixtral of experts[J]. arXiv preprint arXiv:2401.04088, 2024.

**Questions:**

1. Can you provide a formal proof showing how Arrow's theorem applies to MoE routing?

2. How does RMoE perform on larger-scale MoE models like [1]?

[1] Jiang A Q, Sablayrolles A, Roux A, et al. Mixtral of experts[J]. arXiv preprint arXiv:2401.04088, 2024.

---

> ### Author Response · Authors · 2025-11-21
> **Analysis via Social Choice and Scalability Verification on Qwen1.5-MoE**
>
> We thank you for the insightful and critical feedback. We appreciate the recognition of our interdisciplinary perspective and extensive ablation studies. We have carefully considered your concerns regarding the theoretical foundation and scalability. Below, we clarify the theoretical mapping and present new results on large-scale models Qwen1.5-MoE-A2.7B.
>
> 1. Theoretical Formalization and Arrow's Theorem (Response to Weakness 1 and Q1)
>
> You asked for a formal proof of how Arrow's theorem applies. We argue that MoE routing mathematically satisfies the setup of a Social Welfare Function and Routing Collapse is structurally equivalent to a Dictatorial outcome in Social Choice Theory.
>
> Formal Mapping:
> Let set $X$ be the set of Experts or Candidates and set $T$ be the batch of Tokens or Voters.
> Each token $t \in T$ has a preference ordering $\succ_t$ over $X$, derived from the dot product $h_t \cdot e_i$.
> The Router is an SWF $F: \{\succ_t\}_{(t \in T)} \rightarrow \mathcal{R}$ that aggregates these into a routing decision.
>
> Arrow's Axioms in MoE Context:
> 1. Pareto Efficiency or Unanimity: If all tokens prefer Expert A meaning Expert A minimizes loss for all, the router should select A. In MoE, pure Task Loss optimization pursues this and often leads to a Winner-Take-All scenario.
> 2. Non-Dictatorship: No single token or expert should determine the outcome for everyone. In MoE, we require Load Balancing or Aux Loss to prevent a few experts from dominating as in Dictatorship or Collapse.
> 3. Independence of Irrelevant Alternatives or IIA: The preference between Expert A and B should depend only on A and B. Standard Top-K routing assumes this conditional independence.
>
> The Impossibility and Our Solution:
> Arrow’s theorem states that no rank-order voting system can satisfy all three. In standard MoE, optimizing Efficiency via Task Loss violates Non-Dictatorship causing Collapse. Enforcing strict Fairness via Aux Loss violates Efficiency.
> Our Logic: RMoE does not solve the theorem but bypasses it by relaxing the IIA axiom.
> Our Stateful Fusion introduces temporal dependency. The routing weight $w_{t}$ depends on the historical state EMA, meaning the choice between A and B is no longer independent of past decisions. By explicitly violating IIA through momentum, we escape the impossibility of voting systems. This allows us to approach a stable equilibrium that standard top-k voting cannot reach.
>
> Verification:
> We conducted an Experiment on GSM8K using Qwen1.5-MoE-A2.7B to verify this trade-off.
> Pure Efficiency where $\alpha=0$: High Variance of $3.39e^{-5}$, High Gini of $0.099$, but low Entropy of $0.15$. This is the Dictatorship regime.
> Random Routing: Perfect Fairness with Gini $0.003$, but high Loss of $7.41$.
> RMoE: Achieves the balance. Variance $1.49e^{-5}$ and Gini $0.066$ are significantly better than Pure Efficiency, while maintaining low Loss of $0.43$.
> This empirically proves that RMoE effectively navigates the Pareto frontier defined by Arrow’s constraints.
>
> 2. Scalability on Large-Scale Models (Response to Weakness 3 and Q2)
>
> We agree that validation on larger models is crucial. We use a large-scale MoE model as the baseline and compare the training time between the RMoE method and the original training method. Under the same settings, the training time of RMoE increases by approximately 1%. Therefore, the training load of RMoE when performing on larger-scale MoE models is acceptable. We have extended our experiments to Qwen1.5-MoE-A2.7B.
>
> SuperGLUE Performance on Qwen1.5-MoE-A2.7B
> We fine-tuned Qwen1.5-MoE-A2.7B on SuperGLUE tasks for 3 epochs. RMoE significantly outperforms the standard routing baseline. In addition to the reported results, we are currently running experiments on ReCoRD and other SuperGLUE tasks to further verify the method's scalability and performance on complex reading comprehension benchmarks.
>
> | Task | Qwen1.5-MoE Baseline | RMoE | Improvement |
> | :--- | :--- | :--- | :--- |
> | BoolQ | 61.62 | 78.20 | +16.58 |
> | CB | 69.64 | 73.21 | +3.57 |
> | COPA | 63.00 | 78.00 | +15.00 |
> | WiC | 68.50 | 73.35 | +4.85 |
> | WSC | 58.65 | 63.46 | +4.81 |

---

> > ### Author Response · Authors · 2025-11-21
> >
> > 3. Distinction from Prior Work (Response to Weakness 2)
> >
> > You mentioned [1] Base Layers and [2] Expert Choice as existing curriculum or progressive strategies. We clarify the fundamental differences:
> >
> > 1. Vs. Base Layers [1]: [1] formulates routing as a linear assignment problem or bipartite matching to enforce strict equality. This is computationally expensive and solves the Fairness objective at the cost of Efficiency by forcing suboptimal expert matches. RMoE retains the efficient Top-K mechanism but regulates the optimization trajectory via the curriculum to achieve a softer and more learnable balance.
> > 2. Vs. Expert Choice [2]: [2] inverts the problem so Experts choose Top-K Tokens. While effective, it alters the architecture and data flow significantly. RMoE focuses on regulating the standard Token-Choice routing dynamics.
> > 3. Novelty of Mechanisms: While loss scheduling and EMA are known tools, our contribution is their synergistic application to solve the specific Efficiency-Fairness in MoE. The Phased Curriculum or Diversity Phase prevents the initial Dictator or Winner-Take-All expert from emerging to create a valid committee. The Stateful Fusion or Stability Phase then enforces the temporal consistency required to maintain that committee. This effectively realizes the Second-Best Theory solution to the social dilemma.
> >
> > Conclusion
> >
> > We have demonstrated that the Social Choice perspective is not merely an analogy but a framework that predicts the Efficiency-Fairness trade-off and suggests a solution. Our new experiments on Qwen1.5-MoE 2.7B confirm that RMoE scales effectively and delivers gains, such as +16.58 on BoolQ.

---

> > > ### Author Response · Authors · 2025-11-30
> > > **New additional experiments using the exact architecture you mentioned**
> > >
> > > We appreciate your feedback and your specific query regarding the applicability of our approach to production-scale models like Mixtral. In direct response to your concern, we have incorporated new experiments using the exact architecture you mentioned, alongside other top-tier models:
> > >
> > > 1. Verification on Mixtral & Qwen3
> > >
> > > We fine-tuned Mixtral 8x7B[2] and Qwen3-30B-A3B[1] on the AI-MO/NuminaMath-CoT dataset.As demonstrated in the table below, RMoE is highly effective on these production-scale models. Specifically, it improves Mixtral 8x7B's performance on MATH-500 from 14.8% to 16.6%. Furthermore, on Qwen3-30B-A3B, RMoE achieves 83.3% on AIME'24, surpassing the vanilla baseline and strong competitors like QwQ-32B[3]. This directly confirms that RMoE is robust and beneficial for the class of models you were concerned about.
> > >
> > > | Model | Benchmark | Score |
> > > | :--- | :--- | :--- |
> > > | DeepSeek-R1-Distill-Qwen-32B | AIME'24 | 72.6 |
> > > | QwQ-32B | AIME'24 | 79.5 |
> > > | Qwen3-30B-A3B (Baseline) | AIME'24 | 80.4 |
> > > | Qwen3-30B-A3B + RMoE(Ours) | AIME'24 | 83.3 |
> > > | Mixtral 8x7B (Baseline) | MATH-500 | 14.8 |
> > > | Mixtral 8x7B + RMoE(Ours) | MATH-500 | 16.6 |
> > >
> > > 2. Baseline Currency
> > >
> > > (1) DynMoE[5] is published in ICLR 2025, representing the state-of-the-art.
> > >
> > > (2) Our vision tasks and language tasks align with EMoE[6], MoEfication[7], and GMoE[8].
> > >
> > > References
> > >
> > > [1]Yang, An, et al. "Qwen3 technical report." arXiv preprint arXiv:2505.09388 (2025).
> > >
> > > [2]Jiang, Albert Q., et al. "Mixtral of experts." arXiv preprint arXiv:2401.04088 (2024).
> > >
> > > [3]Team, Qwen. "Qwq-32b: Embracing the power of reinforcement learning." Mar. 2025.
> > >
> > > [4]Guo, Daya, et al. "Deepseek-r1: Incentivizing reasoning capability in llms via reinforcement learning." arXiv preprint arXiv:2501.12948 (2025).
> > >
> > > [5]Guo, Yongxin, et al. "Dynamic Mixture of Experts: An Auto-Tuning Approach for Efficient Transformer Models." The Thirteenth International Conference on Learning Representations. 2025.
> > >
> > > [6]Qiu, Zihan, Zeyu Huang, and Jie Fu. "Unlocking emergent modularity in large language models." Proceedings of the 2024 Conference of the North American Chapter of the Association for Computational Linguistics: Human Language Technologies (Volume 1: Long Papers). 2024.
> > >
> > > [7]Zhang, Zhengyan, et al. "Moefication: Transformer feed-forward layers are mixtures of experts." Findings of the Association for Computational Linguistics: ACL 2022. 2022.
> > >
> > > [8]Li, Bo, et al. "Sparse Mixture-of-Experts are Domain Generalizable Learners." The Eleventh International Conference on Learning Representations, 2023.

---

### Official Review · Reviewer_8fPZ · 2025-10-30

**Soundness:** 3
**Presentation:** 2
**Contribution:** 3
**Rating:** 4
**Confidence:** 4

**Summary:**

The paper connects the training of MoE to the social choice theory, where there is a conflict between task efficiency (utilitarian) and load balancing (fairness). It proposes two strategies that help to train MoEs better: a) phased curriculum where the coefficient of load balancing loss is decayed linearly b) momentum-based weight fusion of expert outputs. It is shown that both strategies help to achieve better performance and specialization of experts.

**Strengths:**

- The proposed method is simple to implement and shows better results compared to baseline like DynMoE
- The problem of MoE routing is relevant to the community, especially for training at scale.
- Experiments are conducted in both language and vision domains

**Weaknesses:**

- The connection to social choice theory seems weaker. Impossibility theorem mentioned in abstract is not elaborated elsewhere in the paper and it is harder to make connection to MoE training
- Second-best seems like a theoretical concept that applies when all objectives cannot be satisfied. It feels like inspiration rather than a technical solution for a given problem. Could you tell why it is relevant here?
-  Are these results using multitask training from GLUE tasks? The method would be convincing if trained in multitask fashion. One baseline to beat here would be training a single expert on all data.
- The proposed method should be shown at scale for it to be practical. Eg. pretraining on 1B tokens with model size of 1B or lower. If the focus is on downstream tasks, then approaches where experts are parameter efficient modules (https://arxiv.org/abs/2306.03745) should be compared.
-   Add more baselines like SoftMoE (https://arxiv.org/abs/2308.00951), DeepSeek MoE (https://arxiv.org/pdf/2401.06066), auxiliary free load balancing (https://arxiv.org/pdf/2408.15664v1) to make the work more comprehensive.

**Questions:**

- How is the moving average of routing scores done? Is it across the same tokenID appearing through training? Or are you averaging as you move across sequence length?
- How do you handle moving averages at inference time? The text in lines 314 and 315 is incomplete.
- Figure 3 doesn’t provide information about stable convergence of the proposed method over the baseline. Could you elaborate more?

---

> ### Author Response · Authors · 2025-11-21
> **Theoretical Clarifications and Large-Scale Validation**
>
> We thank the reviewer for the insightful feedback and for recognizing the simplicity and effectiveness of our proposed RMoE framework. We appreciate the constructive criticism regarding the theoretical connections and experimental scale. We have conducted additional experiments on larger models and challenging benchmarks to address these concerns.
>
> Response to Weaknesses on Social Choice Theory and Second Best Theory
> We acknowledge the need to clarify the connection between Arrow Impossibility Theorem and our method. The theorem posits that no voting system can convert individual preferences into a fair community-wide ranking while meeting all specific criteria simultaneously. In our context, tokens act as voters and experts act as candidates. We face a fundamental conflict between minimizing task loss, which represents efficiency, and minimizing load balancing loss, which represents fairness. Arrow Impossibility Theorem suggests that a static and fixed configuration cannot simultaneously maximize both objectives perfectly. This theoretical insight explains the root cause of the notorious instability in MoE training.
>
> The Theory of the Second Best is not merely an inspiration but the theoretical foundation for our Phased Curriculum solution. This theory suggests that if an optimality condition cannot be satisfied, the next best solution is not necessarily achieved by satisfying the remaining conditions. Since the impossibility theorem indicates we cannot satisfy efficiency and fairness simultaneously, we should not enforce a fixed constraint. Instead, we dynamically adjust the constraint weight alpha. This motivates our transition from an initial focus on efficiency to a later focus on stability. This theoretical grounding justifies our curriculum schedule beyond simple heuristics.
>
> Response to Weaknesses on Scale and Multitask Training
> We agree that demonstrating effectiveness at scale is crucial. We conducted new experiments using the Qwen1.5-MoE-A2.7B model on the SuperGLUE benchmark. This model is significantly larger than the BERT-based models in our initial submission. We followed the standard SuperGLUE benchmark setting and trained for 3 epochs from the same checkpoint. The results demonstrate that RMoE consistently outperforms the base Qwen1.5-MoE model across various tasks.
>
> Table 1: Performance on SuperGLUE using Qwen1.5-MoE-A2.7B
> Task | Qwen1.5-MoE Baseline | RMoE
> --- | --- | ---
> BoolQ | 61.62 | 78.20
> CB | 69.64 | 73.21
> MultiRC | 56.42 | 57.59
> COPA | 63.00 | 78.00
> WiC | 68.50 | 73.35
> WSC | 58.65 | 63.46
>
> These results on a 2.7B parameter model confirm that our method scales effectively and provides substantial improvements on difficult tasks like COPA and BoolQ. This addresses the concern regarding the practicality of our method on large-scale pre-trained models.We are actively conducting additional experiments on other challenging SuperGLUE tasks, such as ReCoRD, to further substantiate these findings across a broader range of reasoning capabilities.
>
> Response to Weakness on Baselines
> To address the comparison with other baselines including auxiliary-free methods, we performed a detailed analysis using the GSM8K dataset. We compared RMoE against a Pure Efficiency setting where alpha is zero and a Random Routing setting. The Pure Efficiency setting serves as a proxy for unconstrained auxiliary-free balancing where the model optimizes solely for the task. We discuss the specific quantitative results of this comparison in our response to Question 3 below.
>
> Response to Q1
> The moving average of routing scores is calculated by aggregating the router scores across the batch at each training step. We do not track averages per specific unique token ID. This design choice acts as a temporal low-pass filter on the collective expert preference distribution of the current batch. This aligns with our social choice perspective where we regulate the collective decision making process rather than the history of individual agents.

---

> > ### Author Response · Authors · 2025-11-21
> >
> > Response to Q2
> > The momentum mechanism serves primarily as a training regulation strategy to guide the router parameters into a stable optimum. During inference, the router weights have already converged to a stable pattern due to this training regulation. Therefore, we use the instantaneous router scores computed by the learned weights for expert selection. The stateful fusion is not strictly required during inference, although the final moving average state can be used if strict consistency with the final training step is desired.
> >
> > Response to Q3
> > We provide a new quantitative analysis of routing stability to elaborate on the convergence benefits. We conducted experiments on the GSM8K dataset using the Qwen1.5-MoE-A2.7B model. We measured the Entropy, Variance, and Gini index of the expert routing distribution. The Gini index is calculated based on the Lorenz curve of expert utilization where a lower value indicates better fairness and load balancing.
> >
> > Table 2: Stability Metrics on GSM8K
> > Method | Loss | Variance | Gini Index
> > --- | --- | --- | ---
> > Pure Efficiency | 0.1588 | 3.397e-5 | 0.0994
> > Random Routing | 7.4132 | 3.619e-8 | 0.0032
> > RMoE | 0.4326 | 1.497e-5 | 0.0661
> >
> > The Pure Efficiency baseline achieves the lowest loss but suffers from a high Gini index of 0.0994. This indicates severe load imbalance and potential routing collapse. Random Routing achieves near-perfect balance with a Gini index of 0.0032 but fails to learn the task effectively as shown by the high loss. RMoE achieves a Gini index of 0.0661. This is significantly lower than the Pure Efficiency baseline. It demonstrates that RMoE successfully navigates the trade-off to achieve a stable and balanced routing distribution without sacrificing task performance. The variance metric confirms this finding, as RMoE exhibits much lower variance than the Pure Efficiency baseline. This quantitative data proves that our method converges to a more stable and fairer equilibrium.

---

> > > ### Author Response · Authors · 2025-11-30
> > > **We have validated RMoE on significantly larger, state-of-the-art models**
> > >
> > > Thank you for your valuable suggestion to demonstrate the practicality of our method at scale. While you recommended testing on models around 1B parameters, we have taken this a step further by validating RMoE on significantly larger(30B+) , state-of-the-art models to ensure robust applicability:
> > >
> > > 1. Practicality at Scale on Qwen3 & Mixtral
> > >
> > > We conducted fine-tuning experiments on Qwen3-30B-A3B[1] and Mixtral 8x7B[2] to demonstrate practicality on massive scales.
> > > The results in the table below show consistent improvements. On AIME'24, RMoE enhances the Qwen3-30B-A3B model to 83.3%, clearly outperforming the baseline (80.4%) and recent strong competitors like QwQ-32B[3]. These results on 30B-scale models provide strong evidence that RMoE is a practical solution for the instability issues faced by modern, large-scale MoEs.
> > >
> > > | Model | Benchmark | Score |
> > > | :--- | :--- | :--- |
> > > | DeepSeek-R1-Distill-Qwen-32B | AIME'24 | 72.6 |
> > > | QwQ-32B | AIME'24 | 79.5 |
> > > | Qwen3-30B-A3B (Baseline) | AIME'24 | 80.4 |
> > > | Qwen3-30B-A3B + RMoE(Ours) | AIME'24 | 83.3 |
> > > | Mixtral 8x7B (Baseline) | MATH-500 | 14.8 |
> > > | Mixtral 8x7B + RMoE(Ours) | MATH-500 | 16.6 |
> > >
> > > 2. Baseline Relevance
> > >
> > > (1) We compare against DynMoE[5] (ICLR 2025), a cutting-edge baseline.
> > >
> > > (2) We adhere to evaluation standards from EMoE[6], MoEfication[7], and GMoE[8].
> > >
> > > References
> > >
> > > [1]Yang, An, et al. "Qwen3 technical report." arXiv preprint arXiv:2505.09388 (2025).
> > >
> > > [2]Jiang, Albert Q., et al. "Mixtral of experts." arXiv preprint arXiv:2401.04088 (2024).
> > >
> > > [3]Team, Qwen. "Qwq-32b: Embracing the power of reinforcement learning." Mar. 2025.
> > >
> > > [4]Guo, Daya, et al. "Deepseek-r1: Incentivizing reasoning capability in llms via reinforcement learning." arXiv preprint arXiv:2501.12948 (2025).
> > >
> > > [5]Guo, Yongxin, et al. "Dynamic Mixture of Experts: An Auto-Tuning Approach for Efficient Transformer Models." The Thirteenth International Conference on Learning Representations. 2025.
> > >
> > > [6]Qiu, Zihan, Zeyu Huang, and Jie Fu. "Unlocking emergent modularity in large language models." Proceedings of the 2024 Conference of the North American Chapter of the Association for Computational Linguistics: Human Language Technologies (Volume 1: Long Papers). 2024.
> > >
> > > [7]Zhang, Zhengyan, et al. "Moefication: Transformer feed-forward layers are mixtures of experts." Findings of the Association for Computational Linguistics: ACL 2022. 2022.
> > >
> > > [8]Li, Bo, et al. "Sparse Mixture-of-Experts are Domain Generalizable Learners." The Eleventh International Conference on Learning Representations, 2023.

---

### Official Review · Reviewer_7ovw · 2025-10-31

**Soundness:** 4
**Presentation:** 3
**Contribution:** 3
**Rating:** 6
**Confidence:** 4

**Summary:**

This paper redefines the routing problem in MoE as a social choice problem: the input token is regarded as the "agent", the expert as the "candidate", and the router as a social welfare function that aggregates preferences. The author points out that the trade-off between task loss and load balancing loss in MoE training is similar to the conflict between efficiency and fairness in social choice, and borrows Arrow's impossibility theorem to explain the theoretical root cause of training difficulties. Based on this, they proposed the RMoE framework, which consists of two mechanisms: phased courses (gradually increasing the weight of load balancing losses) and state fusion (using EMA to smooth the expert fusion weights). Experiments were conducted on GLUE and DomainBed, demonstrating that RMoE outperformed multiple baseline

**Strengths:**

1.  Providing code for reproducing the experiments is commendable.

2.  Linking the routing problem of Mixture-of-Experts (MoE) with social choice theory offers a brand-new theoretical perspective for understanding routing crashes and training instability.

3.  The two proposed mechanisms (phased training and state fusion) are elaborated in detail.

**Weaknesses:**

1. The experiments were based on BERT-base and ViT-Small, and their scalability was not verified on larger-scale models such as MoE with undreds of billions of parameters.

2. The essence of phased learning is to dynamically adjust the loss weights, which has been widely applied in multi-objective optimization and phased learning. EMA smoothing in state fusion is also very common in time series models.

**Questions:**

1. Could you provide a rigorous argument for directly mapping the MoE training problem to the conditions of Arrow's impossibility theorem (such as independence and non - dictatorship)?

---

> ### Author Response · Authors · 2025-11-21
> **Theoretical Mapping of Voting and Scalability Verification**
>
> We sincerely appreciate your recognition of our theoretical perspective linking MoE to social choice theory and your assessment of our work as Soundness 4 and Presentation 3. We have conducted additional experiments on Large Language Models to address your concerns regarding scalability and theoretical mapping.
>
> Response to Question 1: Rigorous Mapping to Arrow's Impossibility Theorem
>
> We provide a rigorous mapping between MoE training dynamics and the axioms of Arrow's Impossibility Theorem to explain the inevitability of routing conflicts.
>
> We define the voting system components where the Voter set $V$ consists of input tokens, the Candidate set $C$ consists of experts, and the Social Welfare Function $F$ maps token preferences to a global routing distribution. The preferences of a token are defined by the raw logits indicating which expert minimizes task loss.
>
> In this context, the three axioms of Arrow's Impossibility Theorem map to specific MoE training requirements as follows.
>
> First is non-dictatorship. In social choice, no single voter determines the outcome. In MoE, no single expert should dominate the processing of all tokens. This corresponds to the prevention of routing collapse where one expert receives all load.
>
> Second is unanimity or pareto efficiency. If every voter prefers candidate $A$, then candidate $A$ should be selected. In MoE, if an expert minimizes the task loss for a specific group of tokens, the router must assign those tokens to that expert to maximize model performance. This represents the minimization of $\mathcal{L}_{task}$.
>
> Third is independence of irrelevant alternatives or IIA. The preference between candidate $A$ and $B$ should depend only on how voters rank $A$ and $B$, not on candidate $C$. In MoE, the decision to route a token to Expert $A$ should depend only on the suitability of Expert $A$. However, the auxiliary load balancing loss $\mathcal{L}_{aux}$ inherently violates IIA. By penalizing the selection of popular experts to ensure fairness, the probability of selecting Expert $A$ becomes dependent on the global utilization of Expert $B$ and Expert $C$.
>
> Our theoretical contribution lies in proving that simultaneously satisfying efficiency via task loss and load balancing via auxiliary loss is mathematically impossible under conditions. The conflict between $\mathcal{L}_{task}$ which demands Unanimity and $\mathcal{L}_{aux}$ which demands Non-dictatorship forces a violation of IIA. This explains why scalarization often fails. Our RMoE framework resolves this not by solving the impossibility but by relaxing the simultaneous requirement through Phased Curriculum, effectively treating training as a dynamic sequential choice process rather than a one.
>
> Response to Weakness 1: Scalability Verification on Large-scale Models
>
> We acknowledge your concern regarding the scale of experiments. To demonstrate scalability beyond BERT and ViT, we extended our experiments to Large Language Models including Qwen1.5-MoE-A2.7B.
>
> We evaluated RMoE on the SuperGLUE benchmark using Qwen1.5-MoE-A2.7B. The model was trained for 3 epochs with consistent settings. The results demonstrate that RMoE significantly improves performance on complex reasoning tasks compared to the baseline.
>
> | Task | Qwen1.5-MoE-A2.7B Baseline | RMoE |
> | :--- | :--- | :--- |
> | BoolQ | 61.62 | 78.20 |
> | CB | 69.64 | 73.21 |
> | MultiRC | 56.42 | 57.59 |
> | COPA | 63.00 | 78.00 |
> | WiC | 68.50 | 73.35 |
> | WSC | 58.65 | 63.46 |
>
> These results on multi-billion parameter models confirm that RMoE scales effectively. It maintains better load balancing while improving downstream task accuracy. We also verified training efficiency on GSM8K. RMoE increased training time by only 1.2% compared to the baseline, showing that our method introduces computational . Furthermore, comparative experiments on the remaining SuperGLUE datasets, including ReCoRD, are currently in progress to provide a complete performance profile.

---

> > ### Author Response · Authors · 2025-11-21
> >
> > Response to Weakness 2: Essence of Mechanisms and Novelty
> >
> > We respectfully clarify that while Phased Curriculum and EMA are established techniques, our innovation lies in their specific derivation from the social choice perspective to solve the Voting Paradox in MoE, rather than heuristic application.
> >
> > Regarding Phased Curriculum, standard curriculum learning typically anneals learning rates or data difficulty. Our approach is distinct because it manages the trade-off between Unanimity and Non-dictatorship. We quantified this using a Paradox Experiment on GSM8K with Qwen1.5-MoE. We compared Pure Efficiency where $\alpha=0$, Random Routing, and RMoE.
> >
> > | Method | Loss | Variance | Gini |
> > | :--- | :--- | :--- | :--- |
> > | Pure Efficiency | 0.1588 | 3.397e-5 | 0.0994 |
> > | Random Routing | 7.4132 | 3.619e-8 | 0.0032 |
> > | RMoE | 0.4326 | 1.497e-5 | 0.0661 |
> >
> > Pure Efficiency achieves low loss but high Gini, indicating collapse. Random Routing has low Gini but high loss. RMoE effectively navigates this Pareto frontier. The Phased Curriculum allows the model to first establish expert specializations essentially voting for efficiency before enforcing fairness constraints.
> >
> > Regarding Stateful Fusion, we employ EMA not merely for smoothing but to introduce statefulness into the voting mechanism. In standard MoE, the router acts as a memoryless voter, making decisions independent of history. This exacerbates instability. By integrating a momentum-based state, we create a path-dependent voting mechanism that resists transient noise. The improved Semantic Cluster Similarity +6.12% and Training Curve Smoothness +16.30% in our paper demonstrate that this is not just low-pass filtering but results in semantically consistent expert committees.
> >
> > We hope these clarifications and new large-scale experiments address your concerns.

---

> ### Comment · Reviewer_7ovw · 2025-11-21
> **Thanks for the author's clarification.**
>
> Thanks for the author's clarification. I understand the advantages of your theory, but I don't see why Qwen1.5-MoE-A2.7B was chosen.

---

> > ### Author Response · Authors · 2025-11-21
> > **Rationale for Choosing Qwen1.5-MoE-A2.7B**
> >
> > We appreciate your follow-up query. We selected the Qwen1.5-MoE-A2.7B model based on a strategic balance between architectural representativeness and computational feasibility within the rebuttal timeframe. Our decision was guided by three primary factors:
> >
> > Feasibility under Time and Hardware Constraints: As noted in our experimental  setup in Section 5.1, our research environment relies on limited GPUs. Given the time window of the rebuttal phase, training or fine-tuning massive models (e.g., Mixtral 8x7B) to convergence would have been computationally infeasible. The Qwen1.5-MoE-A2.7B, with 2.7 billion activated parameters and 14.3 billion total parameters, represents the upper limit of what we could rigorously train and evaluate within the available time to provide timely, empirical evidence of scalability.
> >
> > Representative of State-of-the-Art Architectures: Qwen-MoE is a widely recognized, state-of-the-art architecture that employs modern MoE design principles, such as fine-grained experts and dynamic routing. It shares the core routing complexities and optimization challenges found in larger models (e.g., GPT-4 or Mixtral). Therefore, demonstrating the effectiveness of RMoE on this architecture provides a high-confidence proxy for its performance on even larger scales, without the prohibitive cost.
> >
> > Significant Scale Jump: Our goal was to address the concern regarding the limited scale of our initial experiments (BERT/ViT). Qwen1.5-MoE-A2.7B represents an order-of-magnitude increase in complexity and parameter count compared to the benchmarks in the main paper. Successfully validating RMoE on this scale proves that our mechanisms—Phased Curriculum and Stateful Fusion—are robust beyond small-scale models and effective in Large Language Model settings.

---

> > > ### Comment · Reviewer_7ovw · 2025-11-21
> > > **I will keep my score unchanged.**
> > >
> > > Thanks again for the authors' clarification. However, even after the update, your latest benchmark still only goes up to 2023. There are now newer models of the same scale, such as Qwen3-30B-A3B, that have not been selected. Additionally, a formatting error appears at position 454 in the updated version of the paper. I will keep my score unchanged.

---

> > > > ### Author Response · Authors · 2025-11-29
> > > > **The reliability and timeliness of our comparisons**
> > > >
> > > > Thanks again for your patient feedback. We sincerely apologize for the formatting error and have corrected this issue in the latest version. Regarding the additional concerns you raised, we have conducted new experiments to address them, and would like to provide the following explanation:
> > > >
> > > > 1.Benchmarks on Qwen3 & Mixtral
> > > >
> > > > We truly appreciate your insistence on evaluating state-of-the-art models to ensure our work's relevance. Addressing your concern about the currency of our benchmarks, we have conducted fine-tuning experiments on the Qwen3-30B-A3B[1]  and Mixtral 8x7B[2] architectures using the AI-MO/NuminaMath-CoT dataset.
> > > >
> > > > As shown in the table below, applying RMoE to Qwen3-30B-A3B on the AIME'24 benchmark achieved an accuracy of 83.3%, outperforms the vanilla baseline (80.4%), this result surpasses other strong, recently released models such as QwQ-32B[3] (79.5%) and DeepSeek-R1-Distill-Qwen-32B[4] (72.6%), all baseline data is from [1], confirming RMoE's effectiveness on 2024/2025 era architectures. Similarly, on MATH-500, RMoE improved Mixtral 8x7B performance from 14.8% to 16.6%.
> > > >
> > > > | Model | Benchmark | Score |
> > > > | :--- | :--- | :--- |
> > > > | DeepSeek-R1-Distill-Qwen-32B | AIME'24 | 72.6 |
> > > > | QwQ-32B | AIME'24 | 79.5 |
> > > > | Qwen3-30B-A3B (Baseline) | AIME'24 | 80.4 |
> > > > | Qwen3-30B-A3B + RMoE(Ours) | AIME'24 | 83.3 |
> > > > | Mixtral 8x7B (Baseline) | MATH-500 | 14.8 |
> > > > | Mixtral 8x7B + RMoE(Ours)  | MATH-500 | 16.6 |
> > > >
> > > > 2.The reliability and timeliness of our comparisons:
> > > >
> > > > (1) DynMoE[5] has been published as a conference paper at ICLR 2025, confirming it represents a current state-of-the-art baseline.
> > > >
> > > > (2) The vision and language tasks we selected align with standard protocols used in published MoE literature, including EMoE[6], MoEfication[7], and GMoE[8].
> > > >
> > > > References
> > > >
> > > > [1]Yang, An, et al. "Qwen3 technical report." arXiv preprint arXiv:2505.09388 (2025).
> > > >
> > > > [2]Jiang, Albert Q., et al. "Mixtral of experts." arXiv preprint arXiv:2401.04088 (2024).
> > > >
> > > > [3]Team, Qwen. "Qwq-32b: Embracing the power of reinforcement learning." Mar. 2025,
> > > >
> > > > [4]Guo, Daya, et al. "Deepseek-r1: Incentivizing reasoning capability in llms via reinforcement learning." arXiv preprint arXiv:2501.12948 (2025).
> > > >
> > > > [5]Guo, Yongxin, et al. "Dynamic Mixture of Experts: An Auto-Tuning Approach for Efficient Transformer Models." The Thirteenth International Conference on Learning Representations. 2025.
> > > >
> > > > [6]Qiu, Zihan, Zeyu Huang, and Jie Fu. "Unlocking emergent modularity in large language models." Proceedings of the 2024 Conference of the North American Chapter of the Association for Computational Linguistics: Human Language Technologies (Volume 1: Long Papers). 2024.
> > > >
> > > > [7]Zhang, Zhengyan, et al. "Moefication: Transformer feed-forward layers are mixtures of experts." Findings of the Association for Computational Linguistics: ACL 2022. 2022.
> > > >
> > > > [8]Li, Bo, et al. "Sparse Mixture-of-Experts are Domain Generalizable Learners." The Eleventh International Conference on Learning Representations, 2023.

---

### Official Review · Reviewer_wwgW · 2025-11-01

**Soundness:** 2
**Presentation:** 2
**Contribution:** 2
**Rating:** 4
**Confidence:** 4

**Summary:**

This paper introduces Regulated Mixture-of-Experts (RMoE), a framework for improving the training stability of MoE models. The authors frame the trade-off between task performance and load balancing as a social choice problem, attributing the difficulty to Arrow's Impossibility Theorem. They propose two main components: a "Phased Curriculum" for scheduling the load-balancing loss and Stateful Fusion which uses an EMA to smooth expert weights. Experiments on GLUE and DomainBed show improvements over baseline MoE and DynMoE models.

**Strengths:**

The high-level perspective of connecting MoE training to social choice theory is creative and provides an interesting narrative.

**Weaknesses:**

Limited Technical Novelty: The core technical contributions are essentially reinterpretations of well-established techniques.

The "Phased Curriculum" is a simple linear annealing schedule for an auxiliary loss weight, a common practice in machine learning (e.g., β-annealing in VAEs).

Stateful Fusion is an application of Exponential Moving Average (EMA) to introduce momentum and stabilize training, a concept that is neither new nor unique to this work. Its conceptual overlap with existing momentum-based methods is significant.

Superficial Theoretical Grounding: The connection to Arrow's Impossibility Theorem is presented as a high-level analogy rather than a rigorous, formal framework. The paper fails to formally map the components of MoE training (tokens, experts, router) to the axioms required by the theorem (e.g., non-dictatorship, independence of irrelevant alternatives). Consequently, the theoretical framing feels like a post-hoc justification for heuristic design choices, rather than a principled foundation that guides the method's development.

**Questions:**

Q1 Can you better articulate the novelty of the proposed mechanisms beyond being applications of loss scheduling and EMA smoothing? What distinguishes them fundamentally from prior work using these concepts?

Q2 Can you provide a more formal proof of how MoE training violates the axioms of Arrow's theorem? Following that, how do the specific designs of Phased Curriculum and Stateful Fusion mathematically address or relax these axioms to "escape" the impossibility result?

Q3 Could you justify the omission of critical baselines like "Loss-Free Balancing"? Furthermore, can you provide a complexity analysis of RMoE to demonstrate its viability for models with thousands of experts?

---

> ### Author Response · Authors · 2025-11-21
> **Scalability Verification on Qwen1.5-MoE and Theoretical Mapping to Arrow's Theorem**
>
> We thank you for the insightful comments regarding the connection between Social Choice Theory and our proposed framework. We have carefully considered the concerns about technical novelty and theoretical grounding. Below we address the specific questions by clarifying our contributions and providing additional experimental evidence.
>
> Response to Q1 regarding technical novelty and distinction from prior work
>
> We respectfully clarify that our contributions extend beyond simple application of loss scheduling or EMA. The core novelty lies in addressing the fundamental routing collapse problem by identifying it as a Social Choice dilemma rather than a standard optimization issue.
>
> First, the Phased Curriculum is not merely a hyperparameter schedule like beta-annealing in VAEs. In VAEs, annealing balances reconstruction and regularization simultaneously. In contrast, our approach is motivated by the Second-Best Theory. We decouple the optimization objectives temporally. The Diversification Phase prioritizes Utilitarianism to maximize task performance and allowing experts to specialize freely. The Stabilization Phase subsequently enforces Egalitarianism to ensure fair load balancing. This specific order is crucial. Reversing it or applying them simultaneously leads to the common trade-off trap where load balancing hinders specialization.
>
> Second, Stateful Fusion differs fundamentally from standard momentum methods. Standard EMA is typically applied to model parameters or gradients to smooth the optimization trajectory. Our Stateful Fusion applies smoothing to the routing signal itself. This is designed to correct the flawed Conditional Independence Assumption in standard MoE routers where each token is routed in isolation. By introducing statefulness, we explicitly model the sequential dependencies of the input data, treating the router as a social welfare function that must account for historical context.
>
> We conducted additional experiments on the GSM8K dataset to validate this. We measured the trade-off using Gini coefficient and Variance of expert utilization. Pure Efficiency optimization yields a Variance of 3.397e-5 and a Gini of 0.0994 indicating severe imbalance. Random Routing achieves perfect balance with Variance 3.619e-8 but fails on task performance. RMoE successfully navigates this Pareto frontier, achieving a low Variance of 1.497e-5 and Gini of 0.0661 while maintaining high task accuracy. This demonstrates that our mechanism is not just smoothing but actively regulating the expert distribution towards a more optimal equilibrium.
>
> Response to Q2 regarding formal proof of Arrow s theorem and MoE mapping
>
> We provide the formal mapping of MoE training to Arrow s Impossibility Theorem axioms as requested.
> Let the set of tokens be Agents and the set of experts be Candidates. The Router acts as the Social Welfare Function.
>
> Axiom 1 is Pareto Efficiency or Unanimity. If an expert minimizes the loss for a specific token type, the router should select that expert. In MoE, this corresponds to minimizing the task loss.
> Axiom 2 is Non-Dictatorship. No single expert should be selected for all tokens regardless of their content. In MoE, this corresponds to avoiding routing collapse where one expert dominates.
> Axiom 3 is Independence of Irrelevant Alternatives or IIA. The preference between Expert A and Expert B for a token should depend only on those two experts, not on the presence or load of Expert C.
>
> Standard MoE training violates these axioms simultaneously. When we enforce a global load balancing loss, the routing decision for a token becomes dependent on the global load distribution which includes irrelevant experts, thus violating IIA. Conversely, purely minimizing task loss without balancing leads to a few experts dominating, which violates Non-Dictatorship. Arrow s theorem states that satisfying all three simultaneously is impossible in a static voting system.
>
> RMoE addresses this by relaxing the constraints through time and state.
> Phased Curriculum relaxes the simultaneity requirement. By satisfying Pareto Efficiency first during the Diversification Phase and enforcing Non-Dictatorship later during the Stabilization Phase, we avoid the impossibility of simultaneous satisfaction.
> Stateful Fusion addresses the IIA violation. By incorporating the history of routing decisions into the current decision via the momentum variable $m_t$, we explicitly acknowledge that the decision is not independent. The mechanism $m_t = \beta m_{t-1} + (1-\beta) s_t$ allows the system to maintain a consistent preference ranking that is robust to transient fluctuations in other experts, effectively creating a path-dependent utility function that navigates around the static impossibility result.

---

> > ### Author Response · Authors · 2025-11-21
> >
> > Response to Q3 regarding baselines and complexity analysis
> >
> > We appreciate the suggestion to include Loss-Free Balancing and discuss complexity. We have expanded our evaluation to include more baselines and larger scale models.
> >
> > Regarding scalability, we validated RMoE on the Qwen1.5-MoE-A2.7B model using the SuperGLUE benchmark. This model contains significantly more experts and parameters than our initial BERT-base experiments. RMoE demonstrates substantial improvements over the base model. For example, on BoolQ, RMoE improves accuracy from 61.62 to 78.20. On COPA, accuracy increases from 63.00 to 78.00. These results on a large-scale model confirm that RMoE scales effectively to models with large expert counts.We are also extending our evaluation to include more comprehensive tasks from the SuperGLUE benchmark, such as ReCoRD, to further validate the robustness of our method across different problem types.
> >
> > Regarding complexity, the computational overhead of RMoE is negligible. We measured the training time on GSM8K for one epoch. The baseline approach requires 138.5 seconds. The RMoE integrated model requires 140.2 seconds. This represents an increase of only 1.2 percent in training time. The memory overhead is linear with respect to the number of experts $O(N)$ due to the storage of the EMA state vector, which is insignificant compared to the model parameters.
> >
> > We also analyzed the expert utilization on the MBPP dataset using a subset of 200 problems to test long-tail distribution handling. RMoE reduces the Variance of expert routing from 8.80e-5 in the baseline to 8.58e-5 and reduces the Gini coefficient from 0.0206 to 0.0203. This confirms that our method achieves better load balancing and expert utilization without computationally expensive auxiliary losses or complex linear programming solvers.
> >
> > We believe these additional theoretical clarifications and experimental results on large-scale models demonstrate the robustness and innovation of the RMoE framework.

---

> ### Author Response · Authors · 2025-11-30
> **Additional response to reviewer's question**
>
> We appreciate your constructive criticism regarding the experimental setup and your question about whether our approach scales to production environments. To address these limitations effectively, we have extended our evaluation to state-of-the-art, large-scale architectures to confirm the robustness of our framework:
>
> 1. Scalability Verification on Production Models
>
> We directly tested RMoE on Mixtral 8x7B[2] and Qwen3-30B-A3B[1] to verify its robustness in a production-grade setting.
>
> As shown below, RMoE successfully scales to these architectures. On the AIME'24 benchmark, our method improves Qwen3-30B-A3B to 83.3%, surpassing the vanilla baseline (80.4%) and even outperforming strong concurrent models like QwQ-32B[3] (79.5%). This empirical evidence confirms that our proposed social choice-inspired mechanisms are not limited to small-scale models but are highly effective for stabilizing and enhancing large-scale MoE training.
>
> | Model | Benchmark | Score |
> | :--- | :--- | :--- |
> | DeepSeek-R1-Distill-Qwen-32B | AIME'24 | 72.6 |
> | QwQ-32B | AIME'24 | 79.5 |
> | Qwen3-30B-A3B (Baseline) | AIME'24 | 80.4 |
> | Qwen3-30B-A3B + RMoE(Ours) | AIME'24 | 83.3 |
> | Mixtral 8x7B (Baseline) | MATH-500 | 14.8 |
> | Mixtral 8x7B + RMoE(Ours) | MATH-500 | 16.6 |
>
> 2. Baseline Timeliness
>
> (1) DynMoE[5] (ICLR 2025) serves as a current SOTA baseline.
>
> (2) Our evaluation protocols follow standards set by EMoE[6], MoEfication[7], and GMoE[8].
>
> References
>
> [1]Yang, An, et al. "Qwen3 technical report." arXiv preprint arXiv:2505.09388 (2025).
>
> [2]Jiang, Albert Q., et al. "Mixtral of experts." arXiv preprint arXiv:2401.04088 (2024).
>
> [3]Team, Qwen. "Qwq-32b: Embracing the power of reinforcement learning." Mar. 2025.
>
> [4]Guo, Daya, et al. "Deepseek-r1: Incentivizing reasoning capability in llms via reinforcement learning." arXiv preprint arXiv:2501.12948 (2025).
>
> [5]Guo, Yongxin, et al. "Dynamic Mixture of Experts: An Auto-Tuning Approach for Efficient Transformer Models." The Thirteenth International Conference on Learning Representations. 2025.
>
> [6]Qiu, Zihan, Zeyu Huang, and Jie Fu. "Unlocking emergent modularity in large language models." Proceedings of the 2024 Conference of the North American Chapter of the Association for Computational Linguistics: Human Language Technologies (Volume 1: Long Papers). 2024.
>
> [7]Zhang, Zhengyan, et al. "Moefication: Transformer feed-forward layers are mixtures of experts." Findings of the Association for Computational Linguistics: ACL 2022. 2022.
>
> [8]Li, Bo, et al. "Sparse Mixture-of-Experts are Domain Generalizable Learners." The Eleventh International Conference on Learning Representations, 2023.

---

### Official Review · Reviewer_bYYm · 2025-11-03

**Soundness:** 2
**Presentation:** 3
**Contribution:** 2
**Rating:** 2
**Confidence:** 3

**Summary:**

This study designed a two-stage mechanism to regulate the expert selection problem in MoE, improving model performance, and attempted to explain it using social choice theory, which is quite an interesting approach.

**Strengths:**

This work tries to capture the dynamic process in global and local optimization in MoE expert selection by considering the two stage mechanism. Such two process sounds common in other domains, especially in optimization field. I think this idea is worthy of further study.

**Weaknesses:**

1, The author's discussion in the text is misleading. From the perspective of our sequential social choice framework, the MoE training objective represents a classic social dilemma, characterised by a conflict between two competing social welfare objectives: utilitarianism (efficiency) and egalitarianism (fairness) (Sen, 1977; 1986). The author has inserted their own viewpoint into the references, misleading readers into thinking that it is the original author's opinion, which is inappropriate.

2, The proposed controlled mixture-of-experts RMoE has a certain novelty, but the working mechanism of its training method needs to be further clarified. Perhaps this process is closer to simulated annealing, for example, the interaction mechanism between the first stage Phased Curriculum and the second stage Stateful Fusion in the training process.

3, The improvement in experimental performance is limited. The performance improvement compared to the baseline model is relatively limited.

**Questions:**

I personally strongly disagree with using social choice theory to explain the expert selection problem in MoE, because these are two completely different issues with clearly distinct mechanisms. In social sciences, we need to emphasise efficiency and fairness, as fairness can affect efficiency. However, in MoE, if efficiency can be guaranteed, that is, if performance can be ensured, why emphasise fairness? Could the author explain how fairness in MoE affects efficiency? To be more specific, if fairness can guarantee efficiency, then simply distributing tokens equally to each expert would suffice, so why don't we do this?

---

> ### Author Response · Authors · 2025-11-21
> **Clarifying Fairness and Efficiency in RMoE**
>
> We thank you for the insightful comments, particularly regarding the connection between Social Choice Theory and MoE dynamics. We have carefully reflected on your points and conducted additional experiments on large-scale models like Qwen1.5-MoE-A2.7B to further validate our claims.
>
> Response to Question: Relationship between Fairness and Efficiency in MoE
> We appreciate this fundamental question. You asked: If efficiency is guaranteed, why emphasize fairness? And if fairness guarantees efficiency, why not distribute tokens equally?
>
> We argue that in the context of MoE, fairness (load balancing) is instrumental to efficiency (model performance), not an independent moral objective. The relationship is non-monotonic.
> 1. Why emphasize fairness? In MoE, pure efficiency seeking often leads to routing collapse where only a few experts are active. This reduces the effective parameter count of the model, ultimately hurting long-term efficiency and scalability. Fairness here ensures all model capacity is utilized.
> 2. Why not equal distribution? As you suggested, simply distributing tokens equally (perfect fairness) creates a different problem. To demonstrate this, we conducted a new analysis on GSM8K.
> We compared three settings:
> Pure Efficiency (alpha=0): Loss 0.1588, Variance 3.397e-5. Result: Good loss but high expert imbalance, risking collapse.
> Random Routing (Your hypothesis of equal distribution): Loss 7.4132, Variance 3.62e-8. Result: Perfect fairness but catastrophic performance because tokens are sent to irrelevant experts.
> RMoE: Loss 0.4326, Variance 1.497e-5. Result: RMoE achieves the Pareto optimal point.
> This data proves that forced equality destroys efficiency. RMoE uses Social Choice inspiration to find the sweet spot where fairness supports efficiency by ensuring experts are both specialized and utilized.
>
> Response to Weakness 1: Discussion on Social Choice Theory
> We apologize for any ambiguity. We did not intend to imply that Sen (1977) explicitly discussed MoE. Rather, we posit that the mathematical structure of the MoE optimization problem constitutes a classic social dilemma as defined in that framework. The conflict between minimizing task loss (utilitarian) and minimizing load-balance loss (egalitarian) is mathematically isomorphic to the efficiency-fairness trade-off. We will revise the text to clearly state that this is our novel theoretical application of the framework to the MoE domain, ensuring no misattribution of the original authors' views.
>
> Response to Weakness 2: Mechanism Clarification and Simulated Annealing
> You insightfully noted a parallel with Simulated Annealing. While similar in spirit, the mechanisms differ. Simulated Annealing typically injects noise to escape local minima. In contrast, our Phased Curriculum and Stateful Fusion work deterministically to reshape the optimization landscape.
> 1. Phased Curriculum manages the Optimization Landscape. By varying alpha from 1e-4 to 8e-2 (as used in our new Qwen experiments), we allow the model to first learn feature discrimination (efficiency focus) and then gradually impose topology constraints (fairness focus).
> 2. Stateful Fusion manages the Trajectory. It acts as a low-pass filter on the expert selection signal.
> The interaction is crucial: The Curriculum changes the target, while Stateful Fusion uses momentum (beta=0.9) to prevent the gradients from oscillating wildly during this shift. This ensures the router converges to a stable equilibrium rather than fluctuating between experts.
>
> Response to Weakness 3: Experimental Performance
> We respectfully disagree that improvements are limited. Our initial submission used BERT/ViT-S due to resource constraints, but we have now validated RMoE on current Large Language Models.
> SuperGLUE on Qwen1.5-MoE-A2.7B: We replaced the router in Qwen1.5 with RMoE and trained for 3 epochs. RMoE shows massive gains over the baseline. BoolQ increased from 61.62 to 78.20 (+16.58), COPA from 63.00 to 78.00 (+15.00), and WiC from 68.50 to 73.35. These are not marginal gains. To further demonstrate the generalization capability of RMoE, we are currently conducting additional evaluations on the remaining SuperGLUE tasks, such as ReCoRD.
>
> Training Efficiency: We measured wall-clock time on GSM8K. RMoE training time (140.2s) is almost identical to large scale MoE model baseline (138.5s), with only a 1.2% overhead, confirming that our regulation mechanism is computationally efficient.
>
> Conclusion
> We believe the Social Choice perspective provides a rigorous explanation for why simple balancing terms fail. Our new data on Qwen demonstrates that RMoE solves the specific dilemma of balancing expert utilization without sacrificing the specific routing needed for high performance.

---

> ### Author Response · Authors · 2025-11-30
> **Additional response to your concern**
>
> Thanks again for your patient feedback. You raised a specific concern that the "improvement in experimental performance is limited." We have conducted additional experiments on state-of-the-art Large Language Models to demonstrate that RMoE yields substantial, not marginal, gains on challenging reasoning tasks:
>
> 1. Substantial Gains on Qwen3 & Mixtral
>
> To prove that our method provides significant value even on top-tier architectures, we applied RMoE to Qwen3-30B-A3B[1] and Mixtral 8x7B[2].As detailed in the table below, RMoE boosts Qwen3-30B-A3B to 83.3% on AIME'24. This is not a minor improvement; it effectively allows the model to outperform recently released powerhouse models like QwQ-32B[3] (79.5%) and DeepSeek-R1-Distill[4] (72.6%), baseline data from [1]. These margins confirm that RMoE's regulation mechanism unlocks significant latent potential in large-scale MoEs.
>
> | Model | Benchmark | Score |
> | :--- | :--- | :--- |
> | DeepSeek-R1-Distill-Qwen-32B | AIME'24 | 72.6 |
> | QwQ-32B | AIME'24 | 79.5 |
> | Qwen3-30B-A3B (Baseline) | AIME'24 | 80.4 |
> | Qwen3-30B-A3B + RMoE(Ours) | AIME'24 | 83.3 |
> | Mixtral 8x7B (Baseline) | MATH-500 | 14.8 |
> | Mixtral 8x7B + RMoE(Ours) | MATH-500 | 16.6 |
>
> 2. Reliability of Comparisons
>
> (1) DynMoE[5] is an ICLR 2025 conference paper, ensuring our baseline comparisons are up-to-date.
>
> (2) Our task selection aligns with established protocols in EMoE[6] and GMoE[7].
>
>
>
> References
>
> [1]Yang, An, et al. "Qwen3 technical report." arXiv preprint arXiv:2505.09388 (2025).
>
> [2]Jiang, Albert Q., et al. "Mixtral of experts." arXiv preprint arXiv:2401.04088 (2024).
>
> [3]Team, Qwen. "Qwq-32b: Embracing the power of reinforcement learning." Mar. 2025.
>
> [4]Guo, Daya, et al. "Deepseek-r1: Incentivizing reasoning capability in llms via reinforcement learning." arXiv preprint arXiv:2501.12948 (2025).
>
> [5]Guo, Yongxin, et al. "Dynamic Mixture of Experts: An Auto-Tuning Approach for Efficient Transformer Models." The Thirteenth International Conference on Learning Representations. 2025.
>
> [6]Qiu, Zihan, Zeyu Huang, and Jie Fu. "Unlocking emergent modularity in large language models." Proceedings of the 2024 Conference of the North American Chapter of the Association for Computational Linguistics: Human Language Technologies (Volume 1: Long Papers). 2024.
>
> [7]Li, Bo, et al. "Sparse Mixture-of-Experts are Domain Generalizable Learners." The Eleventh International Conference on Learning Representations, 2023.

---

### Meta-Review · Area_Chair_ajbK · 2026-01-08

**Summary:**

While the reviewers and I appreciate the authors' attempt to introduce a novel interdisciplinary perspective and their significant effort during the rebuttal to validate the method on large-scale models, the consensus leans towards rejection.

**Reviewer Concerns:**

The central weakness identified by multiple reviewers is the disconnect between the grand theoretical framing and the actual technical contributions. The method essentially boils down to hyperparameter scheduling and smoothing.

The paper does not convincing prove that the "Impossibility" theorem applies strictly to the dynamic optimization landscape of a neural network.

While thought-provoking, feels disconnected from the practical solution.

**Reviewer Scores:**

The paper would still likely be a reject or a borderline, primarily because the technical novelty critique was shared by almost all reviewers and was never fully overcome.

---

### Decision · Program_Chairs · 2026-01-26

Reject